# ERGODIC MEASURE PRESERVING FLOWS

## ABSTRACT

Training probabilistic models with neural network components is intractable in most cases and requires to use approximations such as Markov chain Monte Carlo (MCMC), which is not scalable and requires significant hyper-parameter tuning, or mean-field variational inference (VI), which is biased. While there has been attempts at combining both approaches, the resulting methods have some important limitations in theory and in practice. As an alternative, we propose a novel method which is scalable, like mean-field VI, and, due to its theoretical foundation in ergodic theory, is also asymptotically accurate, like MCMC. We test our method on popular benchmark problems with deep generative models and Bayesian neural networks. Our results show that we can outperform existing approximate inference methods.

## 1 INTRODUCTION

Approximate statistical inference with unnormalised density functions is fundamentally important problem both Bayesian and frequentist inference. In particular, the successes of many sophisticated generative models in machine learning rely on power inference algorithms. Markov chain Monte Carlo (MCMC) methods and variational inference (VI), originally developed in statistics and physics, are two most important approximate inference methods in machine learning, which has been widely used in all kinds of probabilistic models, like latent topic models (Blei et al., 2003), Boltzmann machines (Hinton, 2002; Salakhutdinov & Larochelle, 2010), Bayesian non-parametric models (Neal, 2000; Kurihara et al., 2007). However, they are facing great challenges in the recent research on probabilistic modelling with deep neural networks (NNs). In particular, Bayesian deep neural networks become popular in recent works, because it exploits the Bayesian framework to overcome the overfitting and data demanding problems in deep learning (Neal, 2012). Another interesting research direction is to use deep neural networks in latent variable models to transform the simple latent random variables into complex distribution, which is also known as to Deep Generative Models (DGM). DGMs have been proved to be very powerful generative models.

Inspired by DGMs, many recent works on variational inference adopted NN to construct flexible approximate distributions. In particular, variational autoencoders (Kingma & Welling, 2014) and normalising flows (Rezende & Mohamed, 2015) are two most influential works in this direction. However, due to lack of understanding of the convergence of specific NNs, the research of NN-based inference is focused on engineering the architecture of inference NNs based on experiments and heuristics.

In this work, we propose a novel approximate inference method based on the classic inference theory of MCMC. Our method is inspired by the idea of parallel simulations of MCMC and the recent advances in variational inference with NNs. Like these variational methods, it is straightforward to accelerate the computation of our method using parallelised simulations on Graphical Processing Units. More importantly, with solid theoretical foundations in the theory of MCMC, the proposed method guarantees asymptotic convergence to arbitrary distributions of interest. It is a great advantage over variational inference, because of the approximation bias in variational methods. Our method is also attractive to a wide range of probabilistic models without NNs and Bayesian NNs.

## 2 BACKGROUND

### 2.1 BAYESIAN NEURAL NETWORKS

Given data $\mathcal{D} = \{\mathbf{x}_n, y_n\}_{n=1}^N$ formed by feature vectors $\mathbf{x}_n$ and corresponding scalar targets $y_n$, we can assume that each $y_n$ is obtained as $y_n = f(\mathbf{x}_n; \boldsymbol{\theta}) + \epsilon_n$, where $f(\cdot; \boldsymbol{\theta})$ is the output of a deep neural network with weights $\boldsymbol{\theta}$ and the $\epsilon_n$ are independent noise variables with $\epsilon_n \sim \mathcal{N}(0, \sigma^2)$. This model specifies a likelihood function $p(y_1, \ldots, y_n | \mathbf{x}_1, \ldots, \mathbf{x}_n, \boldsymbol{\theta})$ which can be combined with a Gaussian prior on $\boldsymbol{\theta}$ to obtain a posterior distribution $p(\boldsymbol{\theta} | \mathcal{D})$. Predictions for the $y_\star$ corresponding to a new feature vector $\mathbf{x}_\star$ are then obtained by using the predictive distribution $p(y_\star | \mathbf{x}_\star, \mathcal{D}) = \int p(y_\star | \mathbf{x}_\star, \boldsymbol{\theta}) p(\boldsymbol{\theta} | \mathcal{D}) \, d\boldsymbol{\theta}$. However, integrating with respect to the posterior distribution $p(\boldsymbol{\theta} | \mathcal{D})$ is intractable and approximations have to be performed in practice, with the most popular methods for this being VI and MCMC.

### 2.2 DEEP GENERATIVE MODELS

Generative models extract intrinsic structure from data by making use of latent variables. Let $\mathcal{D}$ be a dataset with $n$ data points $\{\mathbf{x}_n\}_{n=1}^N$. Given a latent representation $\mathbf{z}$, the data point $\mathbf{x}$ is assumed to be sampled from the conditional distribution $p_{\boldsymbol{\theta}}(\mathbf{x} | \mathbf{z})$, which is specified in terms of some parameters $\boldsymbol{\theta}$. This conditional distribution is often refered to as the decoder. Given a prior distribution $p(\mathbf{z})$ over the latent variables, the joint distribution of data and latent variables is $p_{\boldsymbol{\theta}}(\mathbf{x}, \mathbf{z}) = p(\mathbf{z}) p_{\boldsymbol{\theta}}(\mathbf{x} | \mathbf{z})$. The marginal probability of data $\mathbf{x}$ under the model is then $p_{\boldsymbol{\theta}}(\mathbf{x}) = \int p(\mathbf{z}) p_{\boldsymbol{\theta}}(\mathbf{x} | \mathbf{z}) d\mathbf{z}$.

Until recenlty, $p_{\boldsymbol{\theta}}(\mathbf{x} | \mathbf{z})$ was typically specified using a simple distributional family, e.g. generalized linear models (Murphy, 2012; Bishop, 2006). However, more recently, deep generative models (DGMs) use deep neural networks with weights $\boldsymbol{\theta}$ to specify the decoder (Kingma & Welling, 2014; Goodfellow et al., 2014).

Maximum likelihood is a straightforward way to train probabilistic models with latent variables. However, the marginal likelihood $p_{\boldsymbol{\theta}}(\mathbf{x})$ is intractable to compute in DGMs and approximations are needed in practice. As before, the most popular methods for this are based on VI and MCMC. We briefly describe these methods in the following sections.

### 2.3 VARIATIONAL INFERENCE AND NORMALIZING FLOWS

VI approximates a complex posterior distribution with another simpler parametric distribution which is found by optimizing a lower bound on the marginal likelihood. Let the complex posterior be $p_{\boldsymbol{\theta}}(\mathbf{z} | \mathbf{x})$, with associated marginal likelihood $p(\mathbf{x})$, and let $q_{\boldsymbol{\phi}}(\mathbf{z})$, parameterized by $\boldsymbol{\phi}$, be a simpler tractable distribution. The lower bound of the marginal likelihood is then defined as

$$\log p_{\boldsymbol{\theta}}(\mathbf{x}) \geq \mathbb{E}_{q_{\boldsymbol{\phi}}} \left[ \log p_{\boldsymbol{\theta}}(\mathbf{x}, \mathbf{z}) - \log q_{\boldsymbol{\phi}}(\mathbf{z}) \right] , \tag{1}$$

which is often known as the evidence lower bound (ELBO). The more flexible the parametric family $q_{\boldsymbol{\phi}}$, the better the approximation quality to the true posterior and the tighter the value of the ELBO.

Mean-field VI uses a form for $q_{\boldsymbol{\phi}}(\mathbf{z})$ which assumes independence between random variables. This reduces the computational cost of the optimization problem but often leads to poor performance with complex posterior distributions, such as the ones arising in DGMs or in Bayesian neural networks.

Amortization can be used to accelerate convergence and reduce computational cost when multiple inference problems have to be solved simultaneously. The optimization of (1) can be amortized by making $q_{\boldsymbol{\phi}}$ depend explicitly on the data $\mathbf{x}$. In this case, $q_{\boldsymbol{\phi}}(\mathbf{z})$ is replaced with $q_{\boldsymbol{\phi}}(\mathbf{z} | \mathbf{x})$, where $\boldsymbol{\phi}$ are now the weights of a neural network which computes the parameteres of a tractable parametric distribuion on $\mathbf{z}$ from $\mathbf{x}$. In this manner, for any new value of $\mathbf{x}$, we can readily obtain a correspoding variational approximation given by $q_{\boldsymbol{\phi}}(\mathbf{z} | \mathbf{x})$.

Variational auto-encoders (VAEs) (Kingma & Welling, 2014) are DGMs trained by using mean-field VI with a Gaussian parametric distribuion and amortization. Rezende & Mohamed (2015) improve over this method by using a more flexible variational family called normalizing flows (NFs). The NF family is obtained by applying $L$ invertible non-linear transformations $f_1, \ldots, f_L$ to a random variable $\mathbf{z}_0$ with tractable density $q_0(\mathbf{z}_0)$ and exact simulation. The resulting output is a random

variable $\mathbf{z}_L = f_L \circ \cdots \circ f_1(\mathbf{z}_0)$ with density $q_\phi(\mathbf{z}_L|\mathbf{x}) = q_0(\mathbf{z}_0|\mathbf{x}) \prod_{l=1}^{L} |\partial f_l(\mathbf{z}_{l-1})/\partial \mathbf{z}_{l-1}|^{-1}$ and with $\phi$ being the parameters of $f_1, \ldots, f_L$.

Stochastic gradient descent (SGD), in combination with the reparameterisation trick (Kingma & Welling, 2014), can be used for the scalable optimization of the ELBO in VAEs and NFs. However, the main limitation of VAEs and NFs is the bias present in their variational approximations. This bias can be quite high, even in the case of NFs, since the transformations $f_1, \ldots, f_L$ have to be rather simple to ensure invertibility and to reduce computational costs.

## 2.4 Markov Chain Monte Carlo

Markov chain Monte Carlo (MCMC) is an approximate inference method which does not have the aforementioned bias problem. MCMC works by simulating a stationary Markov chain that generates asymptotically unbiased samples from the distributions of interest. Formally, a Markov chain is a sequence of random variables $\mathbf{z}_0, \mathbf{z}_1, \ldots$ in which the transition from state $\mathbf{z}_l$ to $\mathbf{z}_{l+1}$ is defined by the conditional probability distribution of $\mathbf{z}_{l+1}$ given $\mathbf{z}_l$, denoted by $K(\mathbf{z}_l, \mathbf{z}_{l+1})$. Markov chains in MCMC methods have the following strong stationary property: if one state $\mathbf{z}_l$ of the chain follows the stationary distribution $\pi$, so does the next state $\mathbf{z}_{l+1}$. In particular,

$$\pi(\mathbf{z}_{l+1}) = \int \pi(\mathbf{z}_l) K(\mathbf{z}_l, \mathbf{z}_{l+1}) \, d\mathbf{z}_l \, . \tag{2}$$

If $\mathbf{z}_l$ follows a distribution $\pi_l$ that is different from the stationary one, then the distribution of $\mathbf{z}_{l+1}$

$$\pi_{l+1}(\mathbf{z}_{l+1}) = \int \pi_l(\mathbf{z}_l) K(\mathbf{z}_l, \mathbf{z}_{l+1}) \, d\mathbf{z}_l \, , \tag{3}$$

is guaranteed to be closer to $\pi$ than $\pi_l$. This property implies that, with sufficiently many transitions, the distribution $\pi_l$ of $\mathbf{z}_l$ converges to $\pi$ irrespectively of the distribution the initial state $\mathbf{z}_0$.

Despite beign asymptotically unbiased, MCMC methods are less popular than VI for two reasons. First, they are computationally more expensive and second, they typically include hyper-parameters in the Markov kernel $K$ which are highly problem dependent and are hard to tune in practice.

## 3 Ergodic Measure Preserving Flows

In this section, we describe an inference method that combines the strengths of MCMC and VI and avoids their drawbacks. The idea is to use the output distribution of a MCMC chain, given by (3), as the variational distribution and optimize a simple to evaluate objective function for tuning MCMC parameters. Since MCMC converges to the target asymptotically, our variational approximation can be arbitrarily accurate. The result is a computationally efficient method which avoids the bias of parametric approximations and which can do automatic tuning of hyper-parameters.

### 3.1 Definitions

Given the target distribution $\pi$ with unnormalised density function $\pi^*$, we define an approximate distribution $q$ by a mixture of sequential deterministic transformations that preserves the measure $\pi^*$. We call such approximate distributions *measure preserving flows* (MPFs). The transformations preserving a given measure $\pi^*$ are formally defined as follows (Billingsley, 1986).

**Definition 3.1.** Measure Preserving Transformation (MPT). Let $(\Omega, \mathscr{F}, P)$ be a probability space and $\mu$ be a consistent measure with $P$. A mapping $T : \Omega \to \Omega$ is a measure preserving transformation if $T$ is measurable in both the input filed $\mathscr{F}$ and the output field $\mathscr{F}$ and $\mu(A) = \mu(T^{-1}(A))$ for all $A \in \mathscr{F}$. If $T$ is a one-to-one mapping onto $\Omega$, then T preserves $\mu$: $\mu(A) = \mu(T^{-1}TA) = \mu(TA)$.

An example of MPT is any transformation with Jacobian determinant equal to 1, which preserves the Lebesgue measure. In practice, it is straightforward to verify if a transformation $T$ preserves the measure with density function $\pi^*$ with the following conditions:

  (i) Bijection: $T$ is invertible,
  (ii) Preservation of density function: $\pi^*(\mathbf{z}) = \pi^*(T(\mathbf{z}))$ for all $\mathbf{z}$.

(iii) Preservation of base measure: the Jacobian determinant is one if the Lebesgue measure is the base measure.

In probability theory, MPTs are often used within the area of ergodic stochastic processes since many of these processes can be reformulated as a composition of MPTs. In particular, MCMC kernels are MPTs and stationary Markov chains in MCMC are ergodic processes (Robert & Casella, 2005).

The joint probability of states in a MCMC chain is $q(\mathbf{z}_0, \mathbf{z}_1, \ldots, \mathbf{z}_L) = q_0(\mathbf{z}_0) \prod_{l=1}^{L} K(\mathbf{z}_{l-1}, \mathbf{z}_l)$, where $q_0$ is the distribution of the initial state $\mathbf{z}_0$. The density of the last state $\mathbf{z}_L$ is then obtained by integrating out all the previous states:

$$q_L(\mathbf{z}_L) = \int q_0(\mathbf{z}_0) \prod_{l=1}^{L} K(\mathbf{z}_{l-1}, \mathbf{z}_l) \, d\mathbf{z}_0 \, d\mathbf{z}_1 \ldots d\mathbf{z}_{L-1} \,. \tag{4}$$

If the Markov chain is ergodic, $q_L$ converges to the stationary distribution $\pi$ in total variation distance as the length $L$ of the chain increases (Robert & Casella, 2005).

We define a measure preserving flow (MPF) as a representation of (4) in which the kernel $K$ becomes a deterministic transformation $T_{\mathbf{r}} : Z \to Z$ with stochastic auxiliary input $\mathbf{r}$ following distribution $\mu$. We can then define $\mathbf{z}_L$ as the result of applying these deterministic transformations to $\mathbf{z}_0$, that is, $\mathbf{z}_L = T_{\mathbf{r}_L} \circ \cdots \circ T_{\mathbf{r}_1}(\mathbf{z}_0)$. By following the rule of changing variables, it is then straightforward to derive the density of $\mathbf{z}_L$ as

$$q_L(\mathbf{z}_L) = \int q(\mathbf{z}_L, \mathbf{r}_{1:L}) d\mathbf{r}_{1:L} = \int q(\mathbf{z}_0) \mu(\mathbf{r}_{1:L}) \delta(\mathbf{z}_L - T_{\mathbf{r}_L} \circ \cdots \circ T_{\mathbf{r}_1}(\mathbf{z}_0)) \, d\mathbf{z}_0 \, d\mathbf{r}_{1:L} \,, \tag{5}$$

where $\delta$ denotes the Dirac delta function. Note that there is no Jacobian term in (5) because of the preservation of the Lebesgue measure. Because MPFs are equivalent to ergodic Markov chains, the density obtained at the output of an MPF, that is, $q_L$, will converge to the stationary distribution $\pi$ as $L$ increases.

Hamiltonian Monte Carlo (HMC) is one of the most successful MCMC methods, which also can be interpreted as an MPF. Given the target random variable $\mathbf{z} \in \mathbb{R}^d$ with unnormalised density function $\pi^*$, the HMC kernel is essentially applying a deterministic transformation $\mathcal{H}$ to the previous state $\mathbf{z}_i$ with an auxiliary random variable $\mathbf{r} \in \mathbb{R}^n$ with the density function $\mu(\cdot)$. The transformation $\mathcal{H}$ is given by the solution of Hamiltonian dynamics

$$\dot{\mathbf{z}}(t) = \partial_{\mathbf{r}} K(\mathbf{r}), \quad \dot{\mathbf{r}}(t) = -\partial_{\mathbf{z}} U(\mathbf{z}) \,, \tag{6}$$

where $\dot{\mathbf{z}}$ denotes the derivative of $\mathbf{z}$ w.r.t. the time of the dynamics $t$, $U(\mathbf{z}) = -\log \pi^*(\mathbf{z})$ and $K(\mathbf{r}) = -\log \mu(\mathbf{r})$. The preservation of total Hamiltonian energy, $H(\mathbf{z}, \mathbf{r}) = U(\mathbf{z}) + K(\mathbf{r})$, is the characteristic property of Hamiltonian dynamics, which can be easily verified by $\dot{H}(\mathbf{z}, \mathbf{r}) = 0$ using (6). It is straightforward to see that $\mathcal{H}$ preserves the joint measure $\pi(\mathbf{z})\mu(\mathbf{r})$. In particular, $\mathcal{H}$ is a bijective transformation because the dynamics are deterministic and time reversible (Neal, 2010) and the preservation of the Hamiltonian energy implies the preservation of density. Finally, the volume preservation of $\mathcal{H}$ in the space of $(\mathbf{z}, \mathbf{r})$ is a well known property of Hamiltonian dynamics, which can be proved by Liouville's theorem (Leimkuhler & Reich, 2004; Neal, 2010).

We can write the marginal distribution of the last sample generated by HMC as an MPF:

$$q(\mathbf{z}_L) = \int q(\mathbf{z}_L, \mathbf{r}_{1:L}) d\mathbf{r}_{1:L} = \int q(\mathbf{z}_0) \mu(\mathbf{r}) \delta(\mathbf{z}_L - \mathcal{H}_{\mathbf{r}_L} \circ \cdots \circ \mathcal{H}_{\mathbf{r}_1}(\mathbf{z}_0)) \, d\mathbf{z}_0 \, d\mathbf{r}_{1:L} \,. \tag{7}$$

We call the MPF generated by Hamiltonian dynamics a Hamiltonian MPF (HMPF).

## 3.2 Understanding Measure Preserving Conditions

We would like to address a common misunderstanding on the preservation of volume condition stated by (iii) in Section 3.1. Note that we are interested in sampling the random variable $\mathbf{z}$, but the Hamiltonian dynamics preserve the joint measure $\pi(\mathbf{z}, \mathbf{r})$ rather than $\pi(\mathbf{z})$. Following the conditions of MPTs in Section 3.1, it seems necessary to show that, for any specific value of $\mathbf{r}$, any $T_{\mathbf{r}}$ used within an MPF should preserve volume in $\mathbf{z}$ space. However, this is not the case since the measure preservation conditions in the augmented space $(\mathbf{z}, \mathbf{r})$ are enough to guarantee the preservation of the marginal distribution in $\mathbf{z}$ space. Formally, we have the following proposition:

**Proposition 1.** *Let $T : Z \times \mathcal{E} \to Z \times \mathcal{E}$ preserve the distribution $\pi(\mathbf{z}, \mathbf{r})$. Then, if $\mathbf{r}$ is sampled from $\pi(\mathbf{r}) = \int \pi(\mathbf{z}, \mathbf{r}) \, d\mathbf{r}$, the marginal distribution*

$$\pi(\mathbf{z}) = \int \pi(\mathbf{z}, \mathbf{r}) \, d\mathbf{r}$$

*is also preserved by the projection of $T$ in the space of $\mathbf{z}$, that is, by $T_{\mathbf{r}} : Z \to Z$.*

Proposition 1 gives us some insights on the difference between MPFs and normalising flows (NFs). As mentioned earlier, NFs also use a sequence of transformations $T_{\mathbf{r}} : Z \to Z$. However, these do not preserve the distribution of $\mathbf{z}$ and, consequently, they require the computation of Jacobian determinants by the rule of changing variables. By contrast, Proposition 1 implies that, in MPFs, $T_{\mathbf{r}}$ preserves the marginal $\pi(\mathbf{z})$ if $T$ preserves the joint, which means that there is no need to include any Jacobian computations. For this reason, the transformations used in MPFs can be much more complicated than those used in NFs. For example, in Hamiltonian MPFs, for a given $\mathbf{r}$, the Jacobian of $\mathcal{H}_{\mathbf{r}}$ can be very complicated.

### 3.3 VARIATIONAL INFERENCE WITH MPFs

Given an unnormalized posterior distribution $p_{\boldsymbol{\theta}}(\mathbf{x}, \mathbf{z})$ for $\mathbf{z}$, we can construct an MPF that preserves $p_{\boldsymbol{\theta}}(\mathbf{x}, \mathbf{z})\mu(\mathbf{r})$, where $\mu(\mathbf{r})$ is a simple distribution with tractable density and sampling algorithm. Let $T^{\phi_l}$ be the $l$-th transformation in the flow, where $\phi_l$ are hyper-parameters. This transformation maps the state of the flow from $(\mathbf{z}_{l-1}, \mathbf{r}_l)$ to $(\mathbf{z}_l, \mathbf{r}'_l) = T^{\phi_l}(\mathbf{z}_{l-1}, \mathbf{r}_l)$. Similarly, the composition $T^{\phi_L} \circ \cdots \circ T^{\phi_1}(\mathbf{z}_0, \mathbf{r}_{1:L})$, which we denote by $T^{\phi}$, transforms $(\mathbf{z}_0, \mathbf{r}_{1:L})$ to $(\mathbf{z}_L, \mathbf{r}'_{1:L})$. By the preservation of density and the preservation of Lebesgue measure of $T^{\phi}$, as given by conditions (ii) and (iii) in Section 3.1), we have the following equalities

$$p_{\boldsymbol{\theta}}(\mathbf{x}, \mathbf{z}_0) \prod_{l=1}^{L} \mu(\mathbf{r}_l) = p_{\boldsymbol{\theta}}(\mathbf{x}, \mathbf{z}_1)\mu(\mathbf{r}'_1) \prod_{l=2}^{L} \mu(\mathbf{r}_l) = \cdots = p_{\boldsymbol{\theta}}(\mathbf{x}, \mathbf{z}_L) \prod_{l=1}^{L} \mu(\mathbf{r}'_l), \tag{8}$$

$$q_0(\mathbf{z}_0) \prod_{l=1}^{L} \mu(\mathbf{r}_l) = q_L(\mathbf{z}_1, \mathbf{r}'_1; \phi_1) \prod_{l=2}^{L} \mu(\mathbf{r}_l) = \cdots = q_L(\mathbf{z}_L, \mathbf{r}'_1, \mathbf{r}'_2 \ldots, \mathbf{r}'_L; \phi), \tag{9}$$

where $q_0(\mathbf{z}_0)$ is an initial proposal distribution and $\phi = (\phi_1, \ldots, \phi_L)$ are the transformation hyper-parameters. It is important to clarify that, according to (9), the joint density of $\mathbf{z}_L, \mathbf{r}'_1, \mathbf{r}'_2, \ldots, \mathbf{r}'_L$ is known, but the marginal density for these variables is intractable to compute in general.

Following (1), we can obtain the ELBO for the initial proposal distribution $q_0(\mathbf{z}_0)$ as

$$\mathcal{L}(\mathbf{x}; \boldsymbol{\theta}) = \int \log \frac{p_{\boldsymbol{\theta}}(\mathbf{x}, \mathbf{z}_0)}{q_0(\mathbf{z}_0)} q_0(\mathbf{z}_0) \, d\mathbf{z}_0. \tag{10}$$

We call this expression the simple ELBO. We can then multiplying by the density of the auxiliary variables $\mu(\mathbf{r}_{1:L}) = \prod_{l=1}^{L} \mu(\mathbf{r}_l)$ to obtain

$$\mathcal{L}(\mathbf{x}; \boldsymbol{\theta}) = \int \log \frac{p_{\boldsymbol{\theta}}(\mathbf{x}, \mathbf{z}_0) \prod_{l=1}^{L} \mu(\mathbf{r}_l)}{q_0(\mathbf{z}_0) \prod_{l=1}^{L} \mu(\mathbf{r}_l)} q_0(\mathbf{z}_0) \prod_{l=1}^{L} \mu(\mathbf{r}_l) \, d\mathbf{z}_0 \, d\mathbf{r}_{1:L}. \tag{11}$$

We can then replace $(\mathbf{z}_0, \mathbf{r}_{1:L})$ with $(\mathbf{z}_L, \mathbf{r}'_{1:L})$ in (11) by making use of using transformation $T^{\phi}$, (8) and (9). The result is

$$\mathcal{L}(\mathbf{x}; \boldsymbol{\theta}, \phi) = \int \log \frac{p_{\boldsymbol{\theta}}(\mathbf{x}, \mathbf{z}_L) \prod_{l=1}^{L} \mu(\mathbf{r}'_l)}{q_L(\mathbf{z}_L, \mathbf{r}'_{1:L}; \phi)} q_L(\mathbf{z}_L, \mathbf{r}'_{1:L}; \phi) \, d\mathbf{z}_L \, d\mathbf{r}'_{1:L}, \tag{12}$$

where we have omitted the dependence of $(\mathbf{z}_L, \mathbf{r}'_{1:L})$ on $\phi$, since these variables are determined by the hyper-parameters of the MPTs. We call (12) the reparameterised ELBO.

### 3.4 ERGODIC LOWER BOUND AND ERGODIC INFERENCE

The reparameterised ELBO is of limited use, because it can only be as tight as the ELBO with initial proposal distribution $q_0$. This seems to erase the benefits of using an ergodic MPF, which we know

will converge to the target posterior distribution given a sufficiently long flow. To overcome the drawback of the reparameterised ELBO, we propose another ELBO tailored to the MPF framework, which becomes arbitrarily tight as the length of the flow grows. We call such an ELBO ergodic lower bound (ERLBO).

To derive ERLBO, we first rewrite (12) as

$$\mathcal{L}(\mathbf{x}; \boldsymbol{\theta}, \boldsymbol{\phi}) = \int \log \frac{p_{\boldsymbol{\theta}}(\mathbf{x}, \mathbf{z}_L)}{q_L(\mathbf{z}_L; \boldsymbol{\phi})} q_L(\mathbf{z}_L; \boldsymbol{\phi}) \, d\mathbf{z}_L + \int \log \frac{\prod_{l=1}^{L} \mu(\mathbf{r}_l')}{q_L(\mathbf{r}_{1:L}' | \mathbf{z}_L; \boldsymbol{\phi})} q_L(\mathbf{z}_L, \mathbf{r}_{1:L}'; \boldsymbol{\phi}) \, d\mathbf{z}_L \, d\mathbf{r}_{1:L}'$$

$$= \int \log \frac{p_{\boldsymbol{\theta}}(\mathbf{x}, \mathbf{z}_L)}{q_L(\mathbf{z}_L; \boldsymbol{\phi})} q_L(\mathbf{z}_L; \boldsymbol{\phi}) \, d\mathbf{z}_L - D_{\mathrm{KL}}^L . \tag{13}$$

where $D_{\mathrm{KL}}^L$ is the Kullback-Liebler divergence between $q_L(\mathbf{z}_L, \mathbf{r}_{1:L}'; \boldsymbol{\phi})$ and $q_L(\mathbf{z}_L; \boldsymbol{\phi}) \prod_{l=1}^{L} \mu(\mathbf{r}_l')$. It is straightforward to show that the first term on the RHS in (13) is a lower bound of the marginal likelihood by Jensen's inequality. This leads to the ERLBO given by

$$\tilde{\mathcal{L}}(\mathbf{x}; \boldsymbol{\theta}, \boldsymbol{\phi}) = \int \log \frac{p_{\boldsymbol{\theta}}(\mathbf{x}, \mathbf{z}_L)}{q_L(\mathbf{z}_L; \boldsymbol{\phi})} q_L(\mathbf{z}_L; \boldsymbol{\phi}) \, d\mathbf{z}_L = \mathcal{L}(\mathbf{x}; \boldsymbol{\theta}, \boldsymbol{\phi}) + D_{\mathrm{KL}}^L . \tag{14}$$

This is a tighter lower bound than the simple ELBO because the difference between $\tilde{\mathcal{L}}(\mathbf{x}; \boldsymbol{\theta}, \boldsymbol{\phi})$ and $\mathcal{L}(\mathbf{x}; \boldsymbol{\theta})$ in (10) is $D_{\mathrm{KL}}^L \geq 0$. Moreover, the ERLBO can be shown to monotonically increase w.r.t. $L$.

**Proposition 2.** *The lower bound $\tilde{\mathcal{L}}(\mathbf{x}; \boldsymbol{\theta}, \boldsymbol{\phi})$ in (14) becomes tighter and tighter as $L$ increases, that is, $\tilde{\mathcal{L}}(\mathbf{x}; \boldsymbol{\theta}, \boldsymbol{\phi}_{1:L}) \geq \tilde{\mathcal{L}}(\mathbf{x}; \boldsymbol{\theta}, \boldsymbol{\phi}_{1:L-1})$ and the equality holds if and only if $D_{KL}^L = 0$.*

The complete proof is included in appendix.

Recall that $q_L(\mathbf{z}_L; \boldsymbol{\phi})$ is obtained by a sequence of transformations that preserve the probability measure $p(\mathbf{z}|\mathbf{x})$. It is well-known that MCMC chains have a unique invariant distribution and so do the MCMC-equivalent MPFs. Therefore, we know that if $\tilde{\mathcal{L}}(\mathbf{x}; \boldsymbol{\theta}, \boldsymbol{\phi}_{1:L})$ stops growing, $q_L(\mathbf{z}_L; \boldsymbol{\phi})$ must converge to $p(\mathbf{z}|\mathbf{x})$. This is formally described by the following theorem.

**Theorem 1.** *Given an ergodic measure preserving flow with invariant measure $\pi$, the ergodic lower bound $\tilde{\mathcal{L}}(\mathbf{x}; \boldsymbol{\theta}, \boldsymbol{\phi})$ increases in the length of the flow $L$ and becomes an unbiased estimator of the marginal $\log p(\mathbf{x})$ as $L$ increases to infinity.*

We could tune $\boldsymbol{\phi}$ by optimizing the ERLBO. To better understand the values of $\boldsymbol{\phi}$ that would be favored by this optimisation process, we can rewrite the ERLBO by making explicit its dependence on the entropy of $q_L(\mathbf{z}_L; \boldsymbol{\phi})$, which we denote by $\mathrm{H}[q_L(\mathbf{z}_L; \boldsymbol{\phi})] = -\int \log q_L(\mathbf{z}_L; \boldsymbol{\phi}) q_L(\mathbf{z}_L; \boldsymbol{\phi}) \, d\mathbf{z}_L$. In particular,

$$\tilde{\mathcal{L}}(\mathbf{x}; \boldsymbol{\theta}, \boldsymbol{\phi}) = \mathbf{E}_{q_L(\mathbf{z}_L; \boldsymbol{\phi})} \left[ \log p_{\boldsymbol{\theta}}(\mathbf{x}, \mathbf{z}_L) \right] + \mathrm{H}[q_L(\mathbf{z}_L; \boldsymbol{\phi})] . \tag{15}$$

When optimizing this quantity w.r.t. $\boldsymbol{\phi}$, the first term in the RHS will encourage $q_L(\mathbf{z}_L; \boldsymbol{\phi})$ to have high density in regions where $p_{\boldsymbol{\theta}}(\mathbf{x}, \mathbf{z}_L)$ is high, while the second term will favor high entropy solutions and will prevent $q_L(\mathbf{z}_L; \boldsymbol{\phi})$ from converging to a Dirac delta centered at the maximizer of $\log p_{\boldsymbol{\theta}}(\mathbf{x}, \mathbf{z}_L)$. Note that the first term in the RHS of (15) can be easily approximated by Monte Carlo, while the second term is intractable because $q_L(\mathbf{z}_L; \boldsymbol{\phi})$ is not available. However, since $q_L(\mathbf{z}_L; \boldsymbol{\phi})$ converges to $p_{\boldsymbol{\theta}}(\mathbf{z}|\mathbf{x})$ as $L$ increases, we expect the effect of $\mathrm{H}[q_L(\mathbf{z}_L; \boldsymbol{\phi})]$ on $\boldsymbol{\phi}$ to be small and that most of the similarity of $q_L(\mathbf{z}_L; \boldsymbol{\phi})$ to $p_{\boldsymbol{\theta}}(\mathbf{z}|\mathbf{x})$ will be captured by the first term in the RHS of (15). Therefore, we propose to tune $\boldsymbol{\phi}$ by optimizing the tractable objective given by the first term, that is,

$$\tilde{\mathcal{F}}(\mathbf{x}; \boldsymbol{\theta}, \boldsymbol{\phi}) = \mathbf{E}_{q_L(\mathbf{z}_L; \boldsymbol{\phi})} \left[ \log p_{\boldsymbol{\theta}}(\mathbf{x}, \mathbf{z}_L) \right] . \tag{16}$$

If with the initial flow parameter $\boldsymbol{\phi}_0$, the objective $\tilde{\mathcal{F}} < \mathbf{E}_{p_{\boldsymbol{\theta}}(\mathbf{x}, \mathbf{z})} \left[ \log p_{\boldsymbol{\theta}}(\mathbf{x}, \mathbf{z}) \right]$, we expect optimising $\tilde{\mathcal{F}}$ produces faster convergence of $q_L(\mathbf{z}_L; \boldsymbol{\phi})$ towards $p_{\boldsymbol{\theta}}(\mathbf{x}, \mathbf{z})$. Importantly, the model parameters $\boldsymbol{\theta}$ can also be adjusted by optimizing $\tilde{\mathcal{F}}$ because the omitted $\mathrm{H}[q_L(\mathbf{z}_L; \boldsymbol{\phi})]$ does not depend on $\boldsymbol{\theta}$ and, consequently, optimizing $\tilde{\mathcal{F}}(\mathbf{x}; \boldsymbol{\theta}, \boldsymbol{\phi})$ and $\tilde{\mathcal{L}}(\mathbf{x}; \boldsymbol{\theta}, \boldsymbol{\phi})$ w.r.t. $\boldsymbol{\theta}$ are equivalent operations.

### 3.5 IMPLEMENTATION OF HMPFS

The hyper-parameters to be tuned in a HMPF include the parameters of $q_0$ and the transformation parameters for the Hamiltonian simulation, that is, $\boldsymbol{\phi} = (\boldsymbol{\phi}_1, \ldots, \boldsymbol{\phi}_L)$. A natural choice for $q_0$

is multivariate Gaussian with mean $\boldsymbol{\mu} = (\mu_1 \ldots, \mu_d)$ and diagonal covariance matrix with entries $\boldsymbol{\sigma}^2 = (\sigma_1^2, \ldots, \sigma_d^2)$, where $d$ is the dimensionality of the sample space. The most popular algorithm for simulating Hamiltonian dynamics is the vanilla Leapfrog integrator. We refer to the tutorial of Neal (2010) for more detailed description of the implementation of this algorithm. Leapfrog is a numeric integrator that approximates the Hamiltonian dynamics (6) by an iterative procedure with discretized time $\Delta t$, that is,

$$\mathbf{x}(t + \Delta t) = \mathbf{x}(t) + \Delta t \partial_{\mathbf{r}} K(\mathbf{r}(t)), \qquad \mathbf{r}(t + \Delta t) = \mathbf{r}(t) - \Delta t \partial_{\mathbf{x}} U(\mathbf{x}(t)). \qquad (17)$$

For the flow parameters, we consider the total simulation time $T$. Given a fixed number of Leapfrog iterations $m$, the simulation time $T$ can be reparameterized as the time step size $\Delta t = T/m$. Neal (2010) shows that it is possible to use different $\Delta t$ for each dimension of the sample space to improve the quality of Leapfrog. Therefore, we consider the parameters of the $l$-th Hamiltonian simulation in the flow to be $\phi_l = (\Delta t_{l,1}, \ldots, \Delta t_{l,d})$. The pseudo code for ergodic inference with HMPFs is shown in Algorithm 1.

---

**Algorithm 1:** Ergodic Inference on Hamiltonian Measure Preserving Flow.

---

**input** : potential function $U(\mathbf{z}; \mathbf{x}, \boldsymbol{\theta})$, dataset $\mathcal{D}$ and large $L$
**output:** optimal decoder and flow parameters $\boldsymbol{\theta}^*$ and $\boldsymbol{\phi}^*$

initialize $\boldsymbol{\theta}$ and $\boldsymbol{\phi}$;
**while** *not converged* **do**
    $\mathbf{x} \longleftarrow$ sample one data point from $\mathcal{D}$;
    $\mathbf{z}_0 \sim \mathcal{N}(\boldsymbol{\mu}, \text{diag}(\boldsymbol{\sigma}^2))$;
    /* Start of simulation of HMPF                               */
    **for** $l = 1, \ldots, L$ **do**
        $\mathbf{r} \sim \mathcal{N}(0, 1)$;
        $\mathbf{z}_l \longleftarrow \mathcal{H}(\mathbf{z}_{l-1}, \mathbf{r}; U(\mathbf{z}; \mathbf{x}, \boldsymbol{\theta}), \phi_l)$;               /* Leapfrog simulation */
    **end**
    /* End of simulation of HMPF                                 */
    obj $\longleftarrow U(\mathbf{z}_L; \mathbf{x}, \boldsymbol{\theta})$;    /* one sample Monte Carlo Approx. of $\tilde{\mathcal{F}}(\mathbf{x}; \boldsymbol{\theta}, \boldsymbol{\phi})$ */
    $\boldsymbol{\theta} \longleftarrow$ AdamUpdate$(\boldsymbol{\theta}, \partial_{\boldsymbol{\theta}}\text{obj})$;
    $\boldsymbol{\phi} \longleftarrow$ AdamUpdate$(\boldsymbol{\phi}, \partial_{\boldsymbol{\phi}}\text{obj})$;
**end**

---

We do not include any Metropolis-Hastings (MH) correction steps in our method since this is not necessary. The MH steps are included in MCMC methods to ensure asymptotic convergence to the correct target with an unlimited number of transitions. By contrast, MPFs are in the finite-length regime and the main concern is to accelerate the convergence of MPFs to be as close as possible to the target measure. In this setting, it is not immediately clear that MH steps would be helpful. In particular, in Section 4.1 we provide empirical evidence of how HMPFs can converge to the correct target without MH steps.

Due to the composition of the MPTs, computing the gradient can be expensive when the flow is long. To speed up training, we stop the gradient computations when evaluating $-\partial_{\mathbf{z}} U(\mathbf{z})$ in the Leapfrog steps. This trick leads to incorrect gradients. However, we noticed that the optimization was not significantlly affected by this and still worked very well in practice. Finally, note that working with incorrect gradients does not affect the convergence of the flow to the correct target distribution because that is guaranteed by the convergence of the ERLBO as we mentioned in previous section.

## 4 EXPERIMENTS

We provide empirical evidence of HMPFs in three inference tasks. Our goal is to show that HMPFs can provide better approximations than other approximate inference methods.

### 4.1 DEMONSTRATION OF CONVERGENCE

To verify the theoretical results on the convergence of MPFs in Section 3.4, we test HMPFs on 8 bivariate distributions. The full list of benchmark distributions and results are included in the

appendix. Here, we focus on two multimodal benchmarks. The first testing target distribution is a bimodal moon shaped distribution as shown in Figure 2a. We call this target dual moon. Dual moon is one of the benchmarks in normalising flows (Rezende et al., 2014). The second testing target is a mixture of 6 Gaussian distributions placed in a circle. We use 15 Hamiltonian transformations with 5 Leapfrog steps each. The architecture detail of HMPFs [1] can be found in Section D.1. The initial state of MPFs is sampled from a standard Gaussian distribution. The gradient of the objective function is estimated using 1000 samples from HMPFs.

To illustrate the convergence of HMPFs to the target distribution, figures 1c and 1d show histograms of samples as a function of the flow length and the training iterations. To confirm the convergence numerically, we compute the ERLBO using a numeric method for the estimation of the entropy. Plots for the ERLBO and the ground truth log normalization constants are show in figures 1a and 1b

## 4.2 DEEP GENERATIVE MODELS

MNIST is a standard benchmark for testing approximate inference algorithms for training deep generative models. This dataset contains 60,000 grey level $28 \times 28$ images of handwritten digits. For fair comparison with previous work, we use the 10,000 prebinarised MNIST test images from (Burda et al., 2015)[2]. Our benchmark deep generative model is based on the deconvolutional network used by Salimans et al. (2015) for testing Hamiltonian variational inference (HVI). In particular, the decoder $p_{\boldsymbol{\theta}}(\mathbf{x}, \mathbf{z})$ consists of 32 dimensional latent variables $\mathbf{z}$ with isotropic Gaussian prior $p(\mathbf{z}) = \mathcal{N}(\mathbf{0}, \mathbf{I})$ and a deconvolutional network with the architecture from top to bottom including a single fully-connected layer with 500 RELU hidden units, then three deconvolutional layers with $5 \times 5$ filters, [16, 32, 32] feature maps and RELU activations and the final output layer is simply element-wise logistic activation functions. In the convolutional VAE, the encoder network mirrors the architecture of the decoder.

The code for HVI (Salimans et al., 2015) is not publicly available. Nevertheless, we reimplemented their convolutional VAE and were able to reproduce the marginal likelihood reported by Salimans et al. (2015), as shown in Table 1. This verifies that our implementation of the convolutional VAE is correct and that our results are comparable to the ones reported originally by Salimans et al. (2015). We also implemented HVI following Salimans et al. (2015). We used a single hidden layer network with 640 hidden units and RELU activations as the reverse model for the HMC transitions. We also implemented another VI method similar to HVI and called the Hamiltonian variational encoder (HVAE) (Caterini et al., 2018). Unlike HVI, HVAE does not use a reverse model. This method optimizes instead a bound derived from the stationary distribution of reverse momenta. Futhermore, HVAE uses tempering Hamiltonian dynamics that requires additional Jacobian corrections. In our implementation of HVAE, we simply ignore the temperature for computational efficiency.

| Encoders | Training hours | Training Epochs | Test $\log(\mathbf{x})$ | ESS |
|---|---|---|---|---|
| Conv VAE($n_h$=300) (Salimans et al., 2015) | - | - | -83.20 | - |
| HVI(1HMPF-16LF, $n_h$=800) (Salimans et al., 2015) | - | - | -81.94 | - |
| HVAE(1HMPF-20LF, $n_h$ = 300)(Caterini et al., 2018) | - | - | -84.78 | - |
| Conv VAE($n_h$=500) (**Baseline**) | 6.00 | 3000 | -83.57 | 50 |
| HVI(1HMPF-16LF, $n_h$=800) | 6.00 | 360 | -83.68 | 48 |
| HVAE(1HMPF-16LF, $n_h$=500) | 6.00 | 360 | -84.22 | 48 |
| HMPF(30HMPT-5LF, $n_h$=500, no encoder network) | 1.65 | 54 | -83.17 | 48 |
| HMPF(30HMPT-5LF, $n_h$=500, no encoder network) | 3.00 | 100 | -82.76 | 46 |
| HMPF(30HMPT-5LF, $n_h$=500, no encoder network) | 6.00 | 200 | -82.65 | 45 |
| HMPF(30HMPT-5LF, $n_h$=500, no encoder network) | 12.00 | 400 | **-81.43** | 38 |

Table 1: Comparison in terms of compuational efficiency and approximate test log-likelihood. For fair comparison, we implemented the deconvolutional decoder network in (Salimans et al., 2015) to test HVI. In (Salimans et al., 2015), the test likelihood is estimated using importence-weighted samples from the encoder network. In our experiment, we use a more reliable estimation method based on Hamiltonian annealed importance sampling and report the effective sample size (ESS).

---

[1]The code of HMPFs for all three experiments will be available at `https://github.com/firstauthor/hmpfs`

[2]https://github.com/yburda/iwae

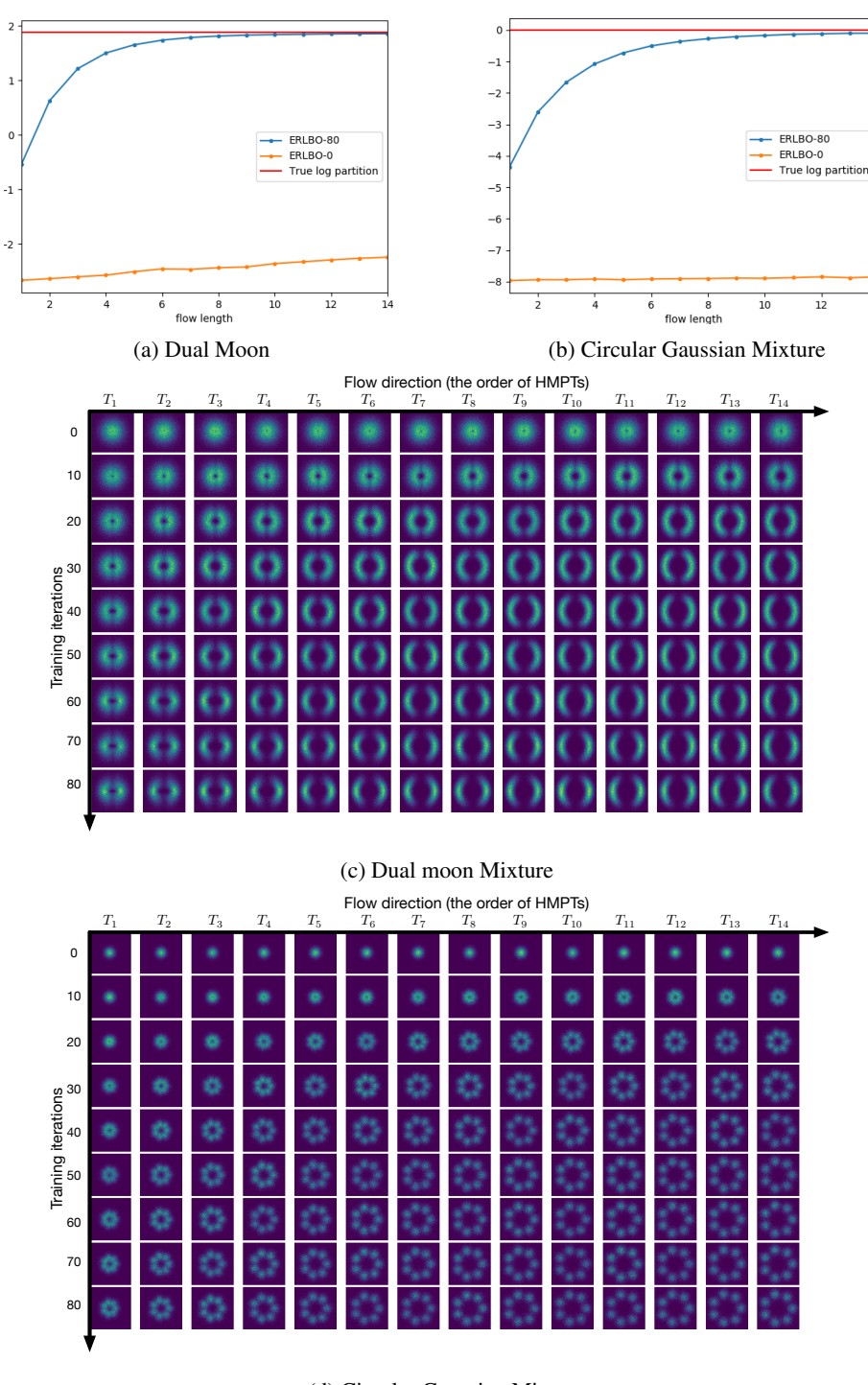

(a) Dual Moon

(b) Circular Gaussian Mixture

(c) Dual moon Mixture

(d) Circular Gaussian Mixture

Figure 1: The demonstration of the convergence of measure preserving flows. Figure (a) and (b) show ergodic lower bounds to the true log normalising constant of ergodic measure preserving flows with 14 transformations. The lower bound is estimated after each transformation as indicated by the axis 'flow length'. The legend 'ERLBO-0' indicates the ERLBO of the flow with the initial randomized flow parameters and the legend 'ERLBO-80' indicates the ERLBO of the flow after 80 training iterations of the flow parameters. Figure (c) and (d) show how the histograms of the 50000 samples from the flow evolve after each transformation (flow direction axis) and every 10 training iterations (training iterations axis).

For HMPF encoder, we use 30 HMPTs with 5 Leapfrog steps per HMPT. The initial distribution $q_0$ is 32 dimensional independent Gaussian. More detailed description of the architecture of HMPFs is in the appendix Section D.1. We optimise the HMPF encoder and the decoder jointly using Adam. The the initial state of the flow is sampled from independent Gaussian. The mean and variance of the initial Gaussian is also optimised jointly with flow and model parameters and their gradients are computed by back propagation given the value of momenta. However, we noticed that with sufficient number of transformations, the effect of optimising initial Gaussian distribution is not significant. Table 1 shows the marginal likelihood of HMPFs and other methods estimated using 100 Hamiltonian annealed importance samples (HAIS) (Wu et al., 2017). We also report the effective sample size (ESS). Overall, HMPFs produce better results and are faster than the baselines.

We also tested the same decoder with HMPFs and convolutional encoder on dynamically binarised Fashion-MNIST (Xiao et al., 2017). The results of test marginal likelihood can be found Table 2.

| Encoders | Training hours | Training Epochs | Test $\log(\mathbf{x})$ | ESS |
|---|---|---|---|---|
| Conv VAE($n_h$=500) | 6.00 | 3000 | -104.90 | 26.3 |
| HMPF(30HMPT-5LF, $n_h$=500, no encoder) | 6.00 | 200 | **-103.087** | 16.2 |

Table 2: The comparison of log marginal likelihood on fashion MNIST between convolutional VAE and HMPFs. We also evaluate HMPFs with different setting of HAIS that gives higher effective sample size (ESS), but the result of test log likelihood is roughly the same.

## 4.3 BAYESIAN NEURAL NETWORKS

In our final experiment we approximate the posterior distribution of Bayesian neural networks. We use four UCI datasets and compare HMPFs with relevant stochastic gradient Hamilton Monte Carlo (SGHMC) methods from (Springenberg et al., 2016). The networks used in these experiments have 50 hidden layers and 1 real valued output unit, as stated in (Springenberg et al., 2016). The HMPFs contain 50 HMC transformation with 3 Leapfrog steps each. The distribution the initial state of the flow is independent Gaussian with mean and variance parameters obtained by fitting a variational Gaussian proposal $q_0$ with Adam optimiser for 200 iterations. To reduce the cost of the Leapfrog iterations, we split training data into 19 mini-batches and only use one random sampled mini-batch for computing the gradient of the potential energy. We train our HMPFs for 10 epochs and the stationary distribution of the flow is chosen as approximate posterior on a random sampled mini-batch. The resulting test log-likelihoods are shown in Table 3. Overall, HMPFs produce significantly better results than SGHMC.

| Method/Dataset | **Boston** | **Yacht** | **Concrete** | **Wine** |
|---|---|---|---|---|
| SGHMC (best average) (Springenberg et al., 2016) | -3.47±0.51 | -13.58±0.98 | -4.87±0.05 | -1.82±0.75 |
| SGHMC (tuned per dataset) (Springenberg et al., 2016) | -2.49±0.15 | -1.75±0.19 | -4.16±0.72 | -1.29±0.28 |
| SGHMC (scale-adapted) (Springenberg et al., 2016) | -2.54±0.04 | -1.11±0.08 | -3.38±0.24 | -1.04±0.17 |
| HMPFs | **-2.17±0.07** | **-0.47±0.06** | **-2.71±0.03** | **-0.71±0.03** |

Table 3: The test log-likelihood of Bayesian neural networks on UCI datasets averaged over 20 splits with 100 sampled network parameters from HMPFs.

## 5 SUMMARY

We have proposed a novel method for approximate inference that combines advantages of variational inference and MCMC methods. We call this method ergodic measure preserving flows (EMPFs). Different from most previous works combining HMC and variational inference, EMPFs enjoy the same asymptotic convergence as HMC and can tune sampling hyper-parameters by optimizing a tractable objective function at a low computational cost. We have shown that EMPFs achieve better results than existing baselines on standard benchmarks. For future work, it will be interesting to study the convergence rate of EMPFs to the target distribution with increasing flow length. Finally, the proposed method can be easily extended to consider recent Riemannian-manifold HMC methods (Zhang & Sutton, 2014; Girolami & Calderhead, 2011) for the construction of the flow.

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

## A   THE PROOF OF PROPOSITION 1

*Proof.* Because $T$ preserves the probability measure $\pi(x, \epsilon)$, then for any measurable set $A$ in Borel set $\mathcal{B}_x \times \mathcal{B}_\epsilon$, we have

$$\pi(A) = \pi(T^{-1}A).$$

Given a set $B \in \mathcal{B}_x$, the set $A_x$ generated by $B \times \mathcal{E}$, is measurable under $\pi$ with the measure

$$\pi(A_x) = \int \pi(B, \epsilon)d\epsilon,$$

that is essentially a probability measure in the space of $x$, also known as the marginal probability

$$\pi(B) = \pi(A_x).$$

Because $T$ preserves the joint measure, applying $T$ on $A_x$ gives

$$TA_x = T(B, \mathcal{E}) = T_\epsilon B \times \mathcal{E},$$

where $T_\epsilon B$ denotes the projection of $TA_x$ in the space of $x$. Follow the definition of measure preserving transformations, we have

$$\pi(A_x) = \int \pi(B, \epsilon)d\epsilon = \int \pi(T_\epsilon^{-1}B, \epsilon)d\epsilon = \pi(T^{-1}A_x),$$

where $T_\epsilon^{-1}$ denotes the preimage of $B$ under $T_\epsilon$. Because $\pi(A_x)$ is essentially the marginal probability $\pi(B)$, $B \in \mathcal{B}_x$, we know the marginal distribution $\pi(x)$ is preserved by $T$ because

$$\pi(B) = \pi(A_x) = \pi(T^{-1}A_x) = \pi(T_\epsilon^{-1}B).$$

This implies that the marginal distribution $\pi(x)$ is preserved by the stochastic mapping

$$T_\epsilon : X \to X, \quad \epsilon \sim \pi(\epsilon),$$

where $\pi(\epsilon) = \int \pi(x, \epsilon)dx$. If $T_\epsilon$ is invertible for any $\epsilon$, we have

$$\pi(B) = \int \pi(B, \epsilon)d\epsilon = \int \pi(T_\epsilon B, \epsilon)d\epsilon = \pi(T_\epsilon B),$$

where $T_\epsilon B$ denotes the projection of $TA_x$ in the space of $x$. Therefore, we know that if we sample $\epsilon$ from $\pi(x, \epsilon)$ and apply $T_\epsilon$ on $x$, then the marginal distribution $\pi(x)$ is preserved. □

## B   THE PROOF OF PROPOSITION 2

*Proof.* Let $\mu$ be the probability measure of auxiliary variable $\mathbf{R}$. Following equation 14, the difference of ERLBO in each measure preserving transformation $T_L$ is given by

$$d\mathcal{L}_{\mathbf{z}}(\mathbf{x}; \theta, \phi_{1:L}) = \mathcal{L}_{\mathbf{z}}(\mathbf{x}; \theta, \phi_{1:L}) - \mathcal{L}_{\mathbf{z}}(\mathbf{x}; \theta, \phi_{1:L-1})$$
$$= D_{\mathrm{KL}}\left(Q_L \middle\| \hat{Q}_L\right) - D_{\mathrm{KL}}\left(Q_{L-1} \middle\| \hat{Q}_{L-1}\right), \tag{18}$$

where $Q_L$ denotes $Q_L(\mathbf{Z}_L, \mathbf{R}'_{1:L})$ and $\hat{Q}_L$ denotes $Q_L(\mathbf{Z}_L) \prod_{l=1}^{L} \mu(\mathbf{R}'_l)$. The KL divergence can be written as the integral

$$D_{\mathrm{KL}}\left(Q_L \middle\| \hat{Q}_L\right) = \int \log \frac{q_L(\mathbf{z}_L, \mathbf{r}'_{1:L}; \phi)}{q_L(\mathbf{z}_L; \phi) \prod_{l=1}^{L} \mu(\mathbf{r}'_l(\phi))} q_L(\mathbf{z}_L, \mathbf{r}'_{1:L}; \phi)d\mathbf{z}_L d\mathbf{r}'_{1:L}. \tag{19}$$

Because $(\mathbf{z}_L, \mathbf{r}'_L)$ is generated from $(\mathbf{z}_{L-1}, \mathbf{r}_L)$ by the deterministic transformation $T_L$ that preserves Lebesgue measure in the phase space, we have the equality

$$q_L(\mathbf{z}_L, \mathbf{r}'_{1:L}) = q_{L-1}(\mathbf{z}_{L-1}, \mathbf{r}'_{1:L-1})\mu(\mathbf{r}_L).$$

Using the reparameterisation

$$(\mathbf{z}_L, \mathbf{r}'_L) = T(\mathbf{z}_{L-1}, \mathbf{r}_L),$$

we can rewrite the second KL term in equation 18 as

$$D_{\mathrm{KL}}\left(Q_{L-1}\middle\|\hat{Q}_{L-1}\right) = \int \log \frac{q_{L-1}(\mathbf{z}_{L-1}, \mathbf{r}'_{1:L-1}; \phi)}{q_{L-1}(\mathbf{z}_{L-1}; \phi) \prod_{l=1}^{L-1} \mu(\mathbf{r}'_l(\phi))} q_{L-1}(\mathbf{z}_{L-1}, \mathbf{r}'_{1:L-1}; \phi) d\mathbf{z}_{L-1} d\mathbf{r}'_{1:L-1}$$

$$= \int \log \frac{q_{L-1}(\mathbf{z}_{L-1}, \mathbf{r}'_{1:L-1}; \phi)\mu(\mathbf{r}_L)}{q_{L-1}(\mathbf{z}_{L-1}; \phi)\mu(\mathbf{r}_L) \prod_{l=1}^{L-1} \mu(\mathbf{r}'_l(\phi))} q_{L-1}(\mathbf{z}_{L-1}, \mathbf{r}'_{1:L-1}; \phi)\mu(\mathbf{r}_L) d\mathbf{z}_{L-1} d\mathbf{r}'_{1:L-1} d\mathbf{r}_L$$

$$= \int \log \frac{q_L(\mathbf{z}_L, \mathbf{r}'_{1:L}; \phi)}{q_L(\mathbf{z}_L, \mathbf{r}'_L; \phi) \prod_{l=1}^{L-1} \mu(\mathbf{r}'_l(\phi))} q_L(\mathbf{z}_L, \mathbf{r}'_{1:L}; \phi) d\mathbf{z}_L d\mathbf{r}'_{1:L}, \qquad (20)$$

where $q_L(\mathbf{z}_L, \mathbf{r}'_L; \phi)$ comes from $q_{L-1}(\mathbf{z}_{L-1}; \phi)\mu(\mathbf{r}_L)$ by the rule of changing variables

$$q_{L-1}(\mathbf{z}_{L-1}; \phi)\mu(\mathbf{r}_L) = \int q_{L-1}(\mathbf{z}_{L-1}, \mathbf{r}'_{1:L-1}; \phi)\mu(\mathbf{r}_L) d\mathbf{r}'_{1:L-1}$$

$$= \int q_L(\mathbf{z}_L, \mathbf{r}'_{1:L}; \phi) d\mathbf{r}'_{1:L-1}$$

$$= q_L(\mathbf{z}_L, \mathbf{r}'_L; \phi).$$

Subtract equation 20 from equation 19, then we have the difference in KL as

$$d\mathcal{L}_{\mathbf{z}}(\mathbf{x}; \theta, \phi_{1:L}) = \int \log \frac{\cancel{q_L(\mathbf{z}_L, \mathbf{r}'_{1:L}; \phi)}}{q_L(\mathbf{z}_L; \phi) \prod_{l=1}^{L} \mu(\mathbf{r}'_l(\phi))} q_L(\mathbf{z}_L, \mathbf{r}'_{1:L}; \phi) d\mathbf{z}_L d\mathbf{r}'_{1:L}$$

$$- \int \log \frac{\cancel{q_L(\mathbf{z}_L, \mathbf{r}'_{1:L}; \phi)}}{q_L(\mathbf{z}_L, \mathbf{r}'_L; \phi) \prod_{l=1}^{L-1} \mu(\mathbf{r}'_l(\phi))} q_L(\mathbf{z}_L, \mathbf{r}'_{1:L}; \phi) d\mathbf{z}_L d\mathbf{r}'_{1:L}$$

$$= \int \log \frac{q_L(\mathbf{z}_L, \mathbf{r}'_L; \phi)\cancel{\prod_{l=1}^{L-1} \mu(\mathbf{r}'_l(\phi))}}{q_L(\mathbf{z}_L; \phi)\mu(\mathbf{r}'_L(\phi))\cancel{\prod_{l=1}^{L-1} \mu(\mathbf{r}'_l(\phi))}} q_L(\mathbf{z}_L, \mathbf{r}'_{1:L}; \phi) d\mathbf{z}_L d\mathbf{r}'_{1:L}$$

$$= \int \log \frac{q_L(\mathbf{z}_L, \mathbf{r}'_L; \phi)}{q_L(\mathbf{z}_L; \phi)\mu(\mathbf{r}'_L(\phi))} q_L(\mathbf{z}_L, \mathbf{r}'_{1:L}; \phi) d\mathbf{z}_L d\mathbf{r}'_{1:L}$$

$$= \int \log \frac{q_L(\mathbf{z}_L, \mathbf{r}'_L; \phi)}{q_L(\mathbf{z}_L; \phi)\mu(\mathbf{r}'_L(\phi))} q_L(\mathbf{z}_L, \mathbf{r}'_L; \phi) d\mathbf{z}_L d\mathbf{r}'_L. \qquad (21)$$

From equation 21, it is easy to see that $d\mathcal{L}_{\mathbf{z}}(\mathbf{x}; \theta, \phi_{1:L})$ is essentially KL divergence between $q_L(\mathbf{z}_L, \mathbf{r}'_L; \phi)$ and $q_L(\mathbf{z}_L; \phi)\mu(\mathbf{r}'_L(\phi))$. This verifies that

$$d\mathcal{L}_{\mathbf{z}}(\mathbf{x}; \theta, \phi_{1:L}) = \mathcal{L}_{\mathbf{z}}(\mathbf{x}; \theta, \phi_{1:L}) - \mathcal{L}_{\mathbf{z}}(\mathbf{x}; \theta, \phi_{1:L-1}) \geq 0.$$

This implies that the ergodic lower bound stops increasing if and only if the KL divergence between $q_L(\mathbf{z}_L, \mathbf{r}'_L; \phi)$ and $q_L(\mathbf{z}_L; \phi)\mu(\mathbf{r}'_L(\phi))$ is equal to 0. $\qquad \square$

## C   PROOF OF THEOREM 1

*Proof.* Because the flow is ergodic with invariant distribution $p(\mathbf{z}|\mathbf{x})$, with sufficient many transformations, $q_L(\mathbf{z})$ can converge to $p(\mathbf{z}|\mathbf{x})$ in total variation distance, that is

$$\lim_{L\to\infty} ||q_L(\mathbf{z}) - p(\mathbf{z}|\mathbf{x})||_{\mathrm{TV}} = 0.$$

This is implied by the monotonic convergence of flow marginal $q_L(\mathbf{x})$ to $\pi(\mathbf{x})$ in total variation simply follows monotonic convergence of MCMC chains in total variation distance by Theorem 6.51 and Proposition 6.52 in (Robert & Casella, 2005).

Then, we have

$$\lim_{L\to\infty} \mathcal{L}_{\mathbf{z}}(\mathbf{x}; \theta, \phi_L) = \lim_{L\to\infty} \int \log \frac{p_\theta(\mathbf{x}, \mathbf{z})}{q_L(\mathbf{z}; \phi)} q_L(\mathbf{z}; \phi) d\mathbf{z}$$

$$= \int \log \frac{p_\theta(\mathbf{x}, \mathbf{z})}{q_\infty(\mathbf{z}_\infty; \phi)} q_\infty(\mathbf{z}; \phi) d\mathbf{z}$$

$$= \int \log \frac{p_\theta(\mathbf{x}, \mathbf{z})}{p_\theta(\mathbf{z}|\mathbf{x})} p_\theta(\mathbf{z}|\mathbf{x}) d\mathbf{z}$$

$$= \log p(\mathbf{x}).$$

$\qquad \square$

# D    EXPERIMENTAL RESULTS

## D.1    CONFIGURATION OF HMPFS

The HMPFs in all the experiments share the following common configuration. The initial distribution of HMPF $q_0(\mathbf{x})$ is given by independent Gaussian $\mathcal{N}(\boldsymbol{\mu}, \boldsymbol{\sigma}^2)$, where $\boldsymbol{\sigma}^2 = (\sigma_1^2, \ldots, \sigma_n^2)$ is the vector of variance. We implement Hamiltonian measure preserving transformation (HMPT) $f_{\mathbf{r}}$ using vanilla Leapfrog integrator for simulating Hamiltonian dynamics $\mathcal{H}$ and independent Gaussian momentum variable $\mathbf{r}$. The implementation of Leapfrog algorithm follows the tutorial of Neal (Neal, 2010). The momentum variables in each HMPT are independent and the each momentum variable has different variance. We consider separate step size $\boldsymbol{\epsilon} = (\epsilon_1, \ldots, \epsilon_n)$ for each dimension of $\mathbf{x}$. Neal (Neal, 2010) shows that tuning leapfrog step size per dimension in HMC is equivalent to tuning the variance vector of momentum variables. So, we generate momentum variables from standard normal and assign an independent Leapfrog step size $\epsilon_l$ for each HMPT $f_l$. The number of iterations in Leapfrog integrator is a fixed parameters based manual tuning. We found that 5 to 10 Leapfrog iterations are often good enough. More Leapfrog steps than that do not give better results. This is consistent with the practice of tuning HMC (Neal, 2010). The intuitive explanation to this is that because Hamiltonian dynamics often have strong oscillation, the longer simulation does not lead to further exploration in sample space.

## D.2    THE TRUE HISTOGRAM OF 2D BENCHMARKS

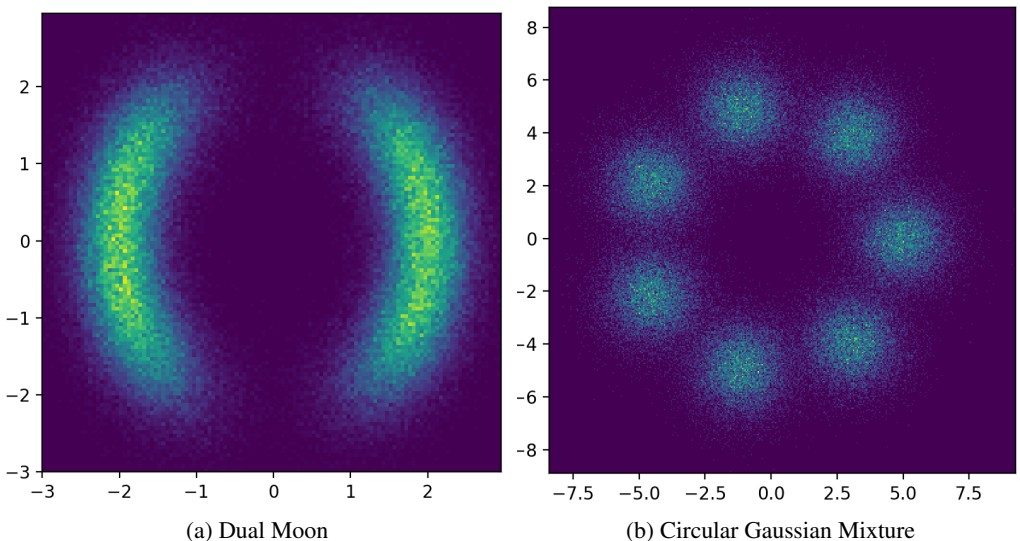

(a) Dual Moon                    (b) Circular Gaussian Mixture

Figure 2: The histogram of perfect samples from the targets using rejection sampling.

## D.3    PLOTS OF MNIST AND FASHION MNIST

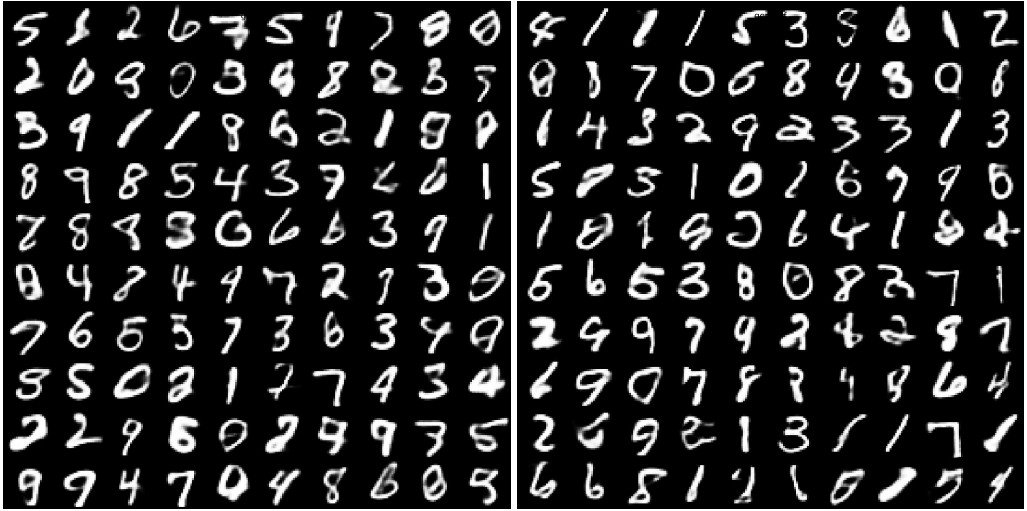

(a) Generation mean image (VAE)   (b) Generation mean image (HMPF)

Figure 3: Random generated images on fashion MNIST. There is no significant visaul difference in the generated images.

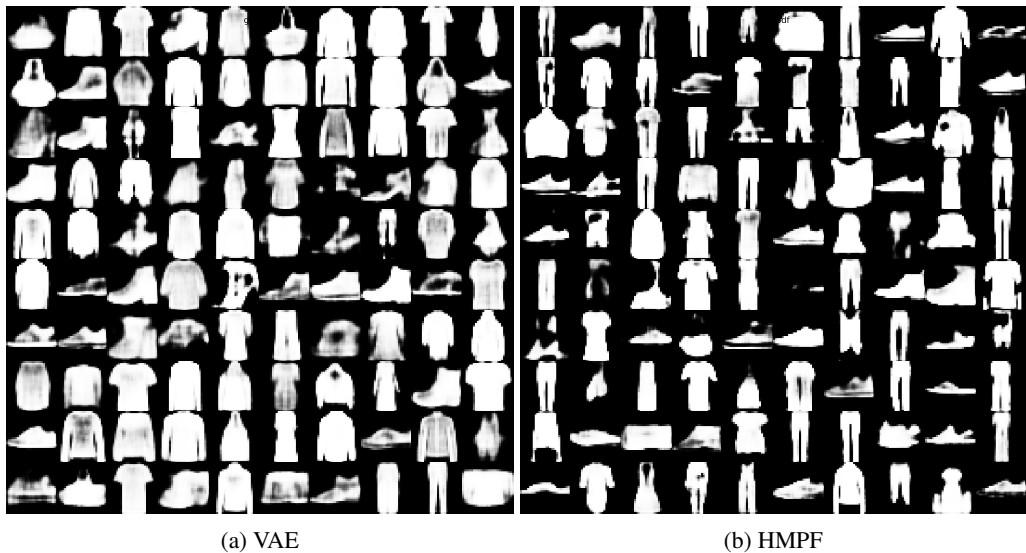

(a) VAE   (b) HMPF

Figure 4: Random generated images on fashion MNIST. It is clear that VAE generates many fashion articles that can almost fill up the whole image, like tops, bags and shirts. In contrast, the generation from the generative model trained using HMPF can generate much diverse products in different size and shapes, like shoes, pants and skirts.

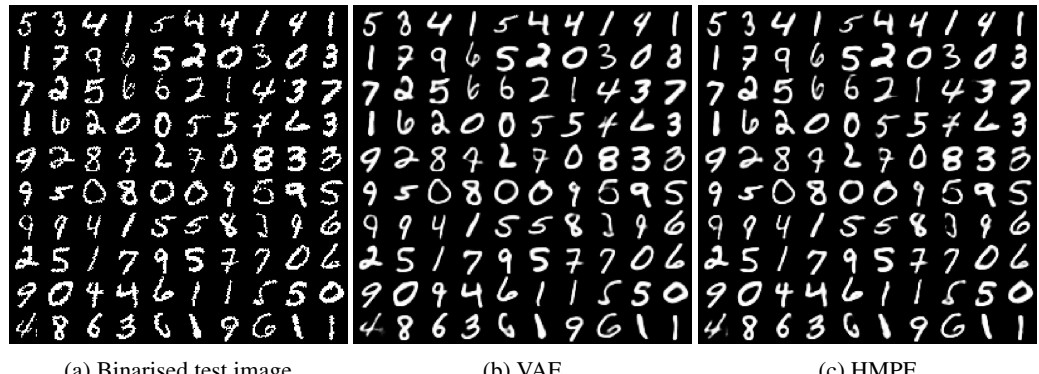

(a) Binarised test image      (b) VAE      (c) HMPF

Figure 5: The reconstructions on MNIST. On the left is dynamically binarised test image. Both convolutional VAE and HMPF reconstruct from the same prebinariesd test image (a) and the generated real valued images are shown as (b) and (c).

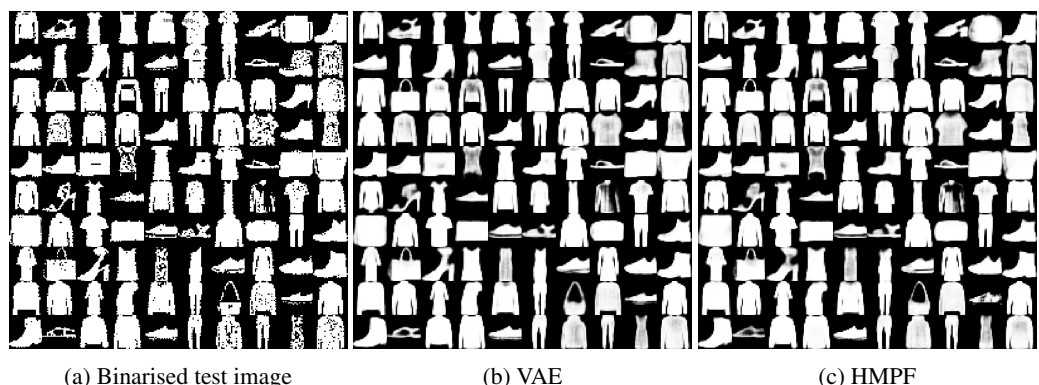

(a) Binarised test image      (b) VAE      (c) HMPF

Figure 6: The reconstructions on fashion MNIST. On the left is dynamically binarised test image. Both convolutional VAE and HMPF reconstruct from the same prebinariesd test image (a) and the generated real valued images are shown as (b) and (c).

