# OpenReview forum: "Ergodic Measure Preserving Flows"
_ICLR.cc/2019/Conference_

### Official Review · AnonReviewer3 · 2018-10-30
**simple idea that might be useful, but unnecessarily complicated exposition**

**Rating:** 4
**Confidence:** 5

**Review:**

This paper proposes training latent variable models (as in VAE decoders) by running HMC to approximate the posterior of the latents, and then estimating model parameters by maximizing the complete data log-likelihood. This is not a new idea by itself and is used e.g. as a baseline in Kingma and Welling's original VAE paper. The novelty in this paper is that it proposes tuning the parameters of the HMC inference algo by maximizing the likelihood achieved by the final sample in the MCMC chain. This seems to work well in practice and might be a useful method, but it is not clear under what conditions it should work.

The paper is written in an unnecessarily complicated and formal way. On first read it seems like the proposed method has much more formal justification than it really has. The discussion up to section 3.5 makes it seem as if there is some new kind of tractable variational bound (the ERLBO) that is optimized, but in practice the actual objective in equation 16 is simply the likelihood at the final sample of the MCMC chain, that is Monte Carlo EM as e.g. used by Kingma & Welling, 2013 as a baseline.  The propositions and theorems seem to apply to an idealized setting, but not to the actual algorithm that is used. They could have been put in an appendix, or even a reference to the exisiting literature on HMC would have sufficed.

The experiments do not clearly demonstrate that the method is much better than previous methods from the literature, although it is much more expensive. (The reported settings require 150 likelihood evaluations per example per minibatch update, versus 1 likelihood evaluation for a VAE). Also see my previous comments about evaluation in this paper's thread.

- Please explain why tuning the HMC algo by maximizing eq 16 should work. I don't think it is a method that generally would work, e.g. if the initial sample z0 ~ q(z|x) is drawn from a data dependent encoder as in HVI (Salimans et al) then I would expect the step size of the HMC to simply go to zero as the encoder gets good. However in your case this does not happen as the initial sample is unconditional from x. Are there general guidelines or guarantees we can conclude from this?

- The authors write "Because MPFs are equivalent to ergodic Markov chains, the density obtained at the output of an MPF, that is, qL, will converge to the stationary distribution π as L increases."

This is true for the idealized flow in continuous time, but HMC with finite step size does generally NOT converge to the correct distribution. This is why practical use of HMC includes a Metropolis-Hastings correction step. You omit this step in your algorithm, with the justification that we don't care about asymptotic convergence in this case. Fair enough, but then you should also omit all statements in the paper that claim that your method converges to the correct posterior in the limit. E.g. the writing makes it seem like Proposition 2 and Theorem 1 apply to your algorithm, but it in fact they do not apply for finite step size. Maybe the statements are still correct if we take the limit with L->inf and the stepsize delta->0 at a certain rate? This is not obvious to me.

In practice, you learn the stepsize delta. Do we have any guarantees this will make delta go to zero at the right rate as we increase the number of steps L? I.e. is this statement from your abstract true? -> "we propose a novel method which is scalable, like mean-field VI, and, due to its theoretical foundation in ergodic theory, is also asymptotically accurate". (convergence of uncorrected HMC only holds in the idealized case with step size -> 0)

---

> ### Author Response · Authors · 2018-11-18
> **Thanks for the review and here is the justifications**
>
> *"The paper is written in an unnecessarily complicated and formal way. On first read it seems like the proposed method has much more formal justification than it really has. The discussion up to section 3.5 makes it seem as if there is some new kind of tractable variational bound (the ERLBO) that is optimized, but in practice the actual objective in equation 16 is simply the likelihood at the final sample of the MCMC chain, that is Monte Carlo EM as e.g. used by Kingma & Welling, 2013 as a baseline.  The propositions and theorems seem to apply to an idealized setting, but not to the actual algorithm that is used. They could have been put in an appendix, or even a reference to the exisiting literature on HMC would have sufficed."
>
> For Eq(8) and Theorem 1, they still hold for a stationary distribution with difference in log density from the target around \epsilon^2, where \epsilon is the leapfrog step size. See reference on this in Radford Neal's tutorial on HMC and "Simulating Hamiltonian Dynamics, Chapter 4, Leimkuhler, Benedict and Reich, Sebastian". Eq (9) holds even without the convergence to target distribution. The reason of that is also very clear in the paper. Radford Neal explains this more clear in his tutorial on HMC.
>
> "Monte Carlo EM" is for training latent variable model rather than optimising kernel parameters in MCMC. In Monte Carlo EM, the expectation is computed by Monte Carlo estimation using perfect or MCMC approximate samples. The Simulation of samples is not optimised or tuned w.r.t. any loss. This is *fundamentally* different from what we propose in this work.
> First, what we propose is an inference method not a training method for latent variable model.  To avoid this confusion, we have the other experiment for Bayesian NNs.
> Second, the same loss function here is the same as likelihood, but we fit the approximation distribution rather than maximum likelihood for latent variable. I hope the reviewer would agree on this: even the same loss function, optimising completely different variable is not a trivial difference.
> Finally, we have the formal condition of ignoring the entropy term: "If with the initial flow parameter φ0, the objective F˜ < Epθ(x,z)[log pθ(x, z)]..." (*right under* the definition of the loss without entropy (equation 16)) For the explanation to this precondition, please check the reply to other reviewers.

---

> > ### Author Response · Authors · 2018-11-18
> > **Follow up**
> >
> > *"- Please explain why tuning the HMC algo by maximizing eq 16 should work. I don't think it is a method that generally would work, e.g. if the initial sample z0 ~ q(z|x) is drawn from a data dependent encoder as in HVI (Salimans et al) then I would expect the step size of the HMC to simply go to zero as the encoder gets good. However in your case this does not happen as the initial sample is unconditional from x. Are there general guidelines or guarantees we can conclude from this?"
> >
> > First of all, let's start with the condition for ergodic Markov chains. A Markov chain is ergodic must satisfy following conditions:
> > -irreducible (precondition of recurrence): The probability of any state change to any other state must be positive.
> > -recurrent: revisit any state with positive probability under stationary infinite many times
> > -aperiodic: the period of the chain must be 1
> > -existing stationary distribution: there must exists a stationary distribution
> >
> > Here is the answer to the question:
> > 1) The irreducibility of ergodic Markov kernel (not in the paper)
> > 2) HMC step size can never be 0 otherwise the kernel is irreducible therefore not ergodic (not in the paper)
> > 3) Our precondition for dropping the entropy term (in the paper, *right below* the definition of the loss without entropy (equation 16))
> >
> > A general guideline to avoid the collapsing of step size to 0 is given by "If with the initial flow parameter φ0, the objective F˜ < Epθ(x,z)[log pθ(x, z)]..." (under the equation (16)) in the paper. This rules out the variational distribution as initial. In particular, because variational approximation will underestimate the variance, it intends to overestimate the the expectation of log p(x, z).

---

> > ### Author Response · Authors · 2018-11-18
> > **Follow up**
> >
> > *In practice, you learn the stepsize delta. Do we have any guarantees this will make delta go to zero at the right rate as we increase the number of steps L? I.e. is this statement from your abstract true? -> "we propose a novel method which is scalable, like mean-field VI, and, due to its theoretical foundation in ergodic theory, is also asymptotically accurate". (convergence of uncorrected HMC only holds in the idealized case with step size -> 0)*
> >
> > Here seems to be a confusion of the definition of discretization step size. Discretization step size can approach 0 under one condition that is the total sum of the step size cannot go to zero with the step size. This is the basic knowledge of numeric methods. I think this explains why the step size cannot be zero in our experiment. We assume the reviewer did not understand the precondition of the loss eq(16), otherwise it is intuitive that the optimal step size cannot be zero.

---

> > > ### Comment · AnonReviewer3 · 2018-11-26
> > > **-**
> > >
> > > this comment is insulting and does not address the question at all

---

> > > > ### Author Response · Authors · 2018-11-26
> > > > **Sincerely Apology for irrelevant reply. Absolutely no intention to offend the reviewer.**
> > > >
> > > > Sorry for the irrelevant reply due to misunderstanding of your question. Sincerely apologise if you are offended by the previous reply. It is absolutely not the intention.
> > > >
> > > > We hope the following reply addressed your question.
> > > >
> > > > "Do we have any guarantees this will make delta go to zero at the right rate as we increase the number of steps L?"
> > > > Yes, we would say so.
> > > > In particular, as metioned explicitly in the paper that
> > > > - the tuning parameter is the total simulation time T.
> > > > - we also define the discretized step size scale is 1/m, m is the number of steps and discretized step size is Delta T = T/m.
> > > > So the Delta T will goes to 0 at the *rate* 1/m as the number of leapfrog steps m goes to infinity.
> > > > Learning Delta T is equivalent to learning T as 1/m is fixed.
> > > >
> > > > "convergence of uncorrected HMC only holds in the idealized case with step size -> 0"
> > > > Totally correct. The proposed method does not converge to the correct target in practice. But as the leapfrog step size is associated with the number of leaps in a reciprocal manner as state earlier. The bias of stationary can be controlled by increasing simulation step size m.  We will adjust the abstract and other points in the paper to avoid the misleading and imprecise statements on asymptotical accurate simulation.
> > > >
> > > > Please accept out apology. Again, no intension to offend you. THANK YOU for the review and many constructive comments!

---

### Official Review · AnonReviewer1 · 2018-11-02
**Interesting idea, missing some baselines and theoretical justifications**

**Rating:** 5
**Confidence:** 3

**Review:**

This paper presents an approximate bayesian inference method based on chaining measure preserving transformation with trainable parameters and optimizing for those using an ad-hoc objective based on a lower-bound on the likelihood.

The paper is clearly written and easy to follow. The proofs seem correct.

In terms of methods, I still have major questions:
- The whole premise of the paper is based on chaining transformations that preserve the target density. However, in practice, you use a leapfrog operator without the Metropolis-Hastings step --what happens to the theoretical guarantees in that case? I'm guessing Eq (8), (9) don't hold anymore and neither does Theorem 1.
- When swapping L for F, could you provide more justifications? You use the argument that p(z|x) ~= q_L so the effect of the entropy term will be negligible. It seems that if they are so similar for large L, why even train the \phi? It also comes back to my first point that in your experiment, the transformations *do not* preserve the target density.
- Regarding the use of measure preserving flow, I think it can be quite hurtful in certain settings -- a very simple example would be a mixture of two gaussians with vastly different variance.

I think this paper also lacks recent references on training parameters for MCMC algorithms, most notably Song et al. (2017) and Levy et al. (2018). Both of these work seem quite related and should be mentioned and compared to. I would have also liked to see the authors contrast their work with Salimans (2015), especially the HVI part; is the main difference the reverse model?

In terms of evaluation, the toy distributions show that the method seems to converge to the right target but does not compare to either vanilla HMC, A-NICE-MC or L2HMC --which all guarantee asymptotic convergence. There should probably also be a mention of one of ESS/Auto-correlation/ESS per sec to get a sense of how helpful the method could be.

For the generative model experiments, I agree with the comments of AnonReviewer3 in that evaluating HMPF-VAE with AIS while evaluating HVI with IWAE is somewhat unfair as the latter can happen to be much looser. I also think a natural baseline to compare to would be Hoffman (2017) or Levy et al. (2018) where after obtaining an approximate posterior sample, these works run an MCMC algorithm before updating the decoder. The algorithms seem to be related (albeit the objectives are slightly different) and should be talked about I think.

References:

Hoffman, Matt. Learning Deep Latent Gaussian Models with Markov Chain Monte Carlo, ICML 2017.

Song, Jiaming et al. A-NICE-MC: Adversarial Training for MCMC, NIPS 2017.

Levy, Daniel et al. Generalizing Hamiltonian Monte Carlo with Neural Networks, ICLR 2018.

---

> ### Author Response · Authors · 2018-11-18
> **Thanks for the review and here is the clarifications**
>
> *"- The whole premise of the paper is based on chaining transformations that preserve the target density. However, in practice, you use a leapfrog operator without the Metropolis-Hastings step --what happens to the theoretical guarantees in that case? I'm guessing Eq (8), (9) don't hold anymore and neither does Theorem 1."
> For Eq(8) and Theorem 1, they still hold for a stationary distribution with difference from the target around \epsilon^2 in log density, where \epsilon is the leapfrog step size. Good reference on this is Radford Neal's tutorial on HMC and "Simulating Hamiltonian Dynamics, Chapter 4, Leimkuhler, Benedict and Reich, Sebastian". Eq (9) holds even without the convergence to target distribution. The reason of that is also very clear in the paper. Radford Neal explains this more clear in his tutorial on HMC.
>
> *"- When swapping L for F, could you provide more justifications? You use the argument that p(z|x) ~= q_L so the effect of the entropy term will be negligible. It seems that if they are so similar for large L, why even train the \phi? It also comes back to my first point that in your experiment, the transformations *do not* preserve the target density. "
>
> We made it clear that the precondition for ignoring the entropy term in the paper: "If with the initial flow parameter φ0, the objective F˜ < Epθ(x,z)[log pθ(x, z)]..." (*right under* the definition of the loss without entropy (equation 16))
> We are happy to make this more obvious somehow. Intuitively, it is not hard to see why this precondition make sense. In particular, we want to maximize this loss to converge to the target, then the initial loss value should be lower than the loss value of the target. This precondition is what it means formally for "p(z|x) ~= q_L so the effect of the entropy term will be negligible". If "p(z|x) ~= q_L" is taken out of the context, in particular, the formal condition we give in the paper, your question makes sense. But, the critical theoretical justification is in the paper. We can work on to make this formal precondition better explained and connected with "p(z|x) ~= q_L".
>
> *"- Regarding the use of measure preserving flow, I think it can be quite hurtful in certain settings -- a very simple example would be a mixture of two gaussians with vastly different variance. "
>
> We guess by this example, the reviewer want to say MCMC can miss some mode in this example. I think what the review want to propose by this example is the classic problem of MCMC of being trapped in high density area, even the probability mass is small. Please confirm this is what you mean.
>
> If so, here is the answer. As long as the ergodicity holds, the convergence to arbitrary distribution holds for any measure preserving flow. Measure preserving is the precondition of ergodicity, so *measure preserving never hurts convergence to the correct target*. However, the target measure is preserved is not sufficient for ergodicity. In particular, many MCMC kernels are weak in irreducibility. On continuous target with the big density gap, many MCMC mehods, including HMC, have difficulties in exploring high probability area with low density. This is not an issue if the precondition "with the initial flow parameter φ0, the objective F˜ < Epθ(x,z)[log pθ(x, z)]..." holds. Because this implies that with initial parameter, the flow distribution should explore the low density area even more than the target distribution.

---

> > ### Author Response · Authors · 2018-11-18
> > **Follow up**
> >
> > *"I think this paper also lacks recent references on training parameters for MCMC algorithms, most notably Song et al. (2017) and Levy et al. (2018). Both of these work seem quite related and should be mentioned and compared to. I would have also liked to see the authors contrast their work with Salimans (2015), especially the HVI part; is the main difference the reverse model?
> >
> > In terms of evaluation, the toy distributions show that the method seems to converge to the right target but does not compare to either vanilla HMC, A-NICE-MC or L2HMC --which all guarantee asymptotic convergence. There should probably also be a mention of one of ESS/Auto-correlation/ESS per sec to get a sense of how helpful the method could be."
> >
> > Yes, many MCMC methods are relevant, but the reviewer seems confused with the nature of the proposed method. The proposed method is to simulate finite-step ergodic Markov chain that transforms i.i.d. samples from q_0 to q_N, therefore the samples from q_N are i.i.d.. We think it is not fair to compare the proposed method with MCMC on ESS/Auto-correlation/ESS per sec. ESS of any i.i.d. samples is equal to the number of samples. Because the simulation can be parallelized, ESS per sec becomes 0 as the number of samples goes infinity. This is the motivation of this work. The review may have different opinion on this. Could you clarify why ESS and auto-correlation related metrics are relevant?
> >
> > *"For the generative model experiments, I agree with the comments of AnonReviewer3 in that evaluating HMPF-VAE with AIS while evaluating HVI with IWAE is somewhat unfair as the latter can happen to be much looser. I also think a natural baseline to compare to would be Hoffman (2017) or Levy et al. (2018) where after obtaining an approximate posterior sample, these works run an MCMC algorithm before updating the decoder. The algorithms seem to be related (albeit the objectives are slightly different) and should be talked about I think."
> >
> > Ok, we also agree with the reviewer on this. How about a simple fix the issue on "unfair comparison of different biased estimators". It is trivial to recover the unbiased estimation of likelihood by taking exponential of the reported log likelihood and our number. Then, the comparison of two unbiased estimator should be fair now. Otherwise, please clarify why it is still unfair to compare two unbiased estimators. (Unless you want to consider the variance of the estimator, but as far as I know there is no literature report the variance of test likelihood.) Moreover, we replicated the experiment in the literature and evaluate the baseline in HAIS. Could you clarify why this is unfair to baseline?

---

> > > ### Comment · AnonReviewer1 · 2018-11-26
> > > **Response**
> > >
> > > I have read the follow-up of the authors and stand by my initial evaluation.
> > >
> > > - A-NICE-MC/L2HMC are still learned kernel one should compare to; even if you consider i.i.d samples after running those kernels for a fix number of steps.
> > > - AIS is known to be a notably tighter lower bound to IWAE and thus using AIS for your model vs IWAE for Salimans (2015) will give you an advantage.
> > > - I also stand by my comment regarding measure-preserving flows; I agree that asymptotic convergence still holds but the probability to move into a low density high probability area can be exponentially small, which is fixed by not constraining to measure-preserving.

---

> > > > ### Author Response · Authors · 2018-11-26
> > > > **Thanks for the response**
> > > >
> > > > - First, please confirm that you understand that the ESS of fixed number of i.i.d. samples is always higher than the same number of samples from any MCMC method. Second, can you explain why you insists to compare with auto-tuning MCMC with ESS and sample correlation?
> > > >
> > > > - We never recommend direct comparison with literature due to implementation difference in model, training and evaluation strategy. I have the results of HVI using HAIS in our paper and that is the one for fair comparison. Can you please explain why the results from Salimans (2015) is relevant for fairness in this case?
> > > >
> > > > - "Move into a low density high probability area can be exponentially small, which is fixed by not constraining to measure-preserving" Do you have formal proof on this for our method? It is straightforward to prove this for MCMC because MCMC is initialized with fixed state and the chain can get stuck in high density area. But, the initial state of our method is random and the initial distribution is guaranteed to cover lower density area under the condition "with the initial flow parameter φ0, the objective F˜ < Epθ(x,z)[log pθ(x, z)]..." This suggests that our method would suffer less from the problem due to the random initialization.

---

### Official Review · AnonReviewer2 · 2018-11-07
**An interesting heuristic and some interesting derivations, but there's a gap between the two.**

**Rating:** 5
**Confidence:** 4

**Review:**

This paper proposes a simple heuristic for tuning HMC's parameters: just optimize the expected log-density of the Lth sample. It seems to work reasonably well on the problems the authors evaluate on.

This heuristic is arrived at by a somewhat roundabout derivation, which I found interesting (although many of the same ideas are implicit in Salimans et al. (2014; "MCMC & VI: Bridging the Gap")). But ultimately this derivation comes to a head at this very heuristic argument:

“…since qL(zL; φ) converges to pθ(z|x) as L increases, we expect the effect of H[qL(zL; φ)] on φ to be small and that most of the similarity of qL(zL; φ) to pθ(z|x) will be captured by the first term in the RHS of (15). Therefore, we propose to tune φ by optimizing the tractable objective given by the first term…”:

I don’t see why this argument applies to the entropy term and not to the log-joint term. In particular, If q_L has really converged to p(z|x), then there’s no point optimizing φ either, right?

Here’s a concrete example of how I could imagine this procedure going wrong: make q(z0) a delta at the latent vector z* that maximizes the log-joint, and set the step size of the Hamiltonian simulation to 0. This will make the entropy term (which is ignored) -∞, maximize the log-joint term, and I think it even makes D^L_{KL}=0.

It seems like this isn’t what actually happens experimentally, though—perhaps I’m missing something?

Regarding the experiments, a natural baseline would be something akin to the approach of Hoffman (2017; "Learning Deep Latent Gaussian Models with Markov Chain Monte Carlo”), simply initializing the HMC sampler with a mean-field Gaussian. I would expect this to produce worse results for small numbers of steps, since the variational Gaussian would choose a single mode, but I’m curious how the quantitative metrics would compare.

Some more minor points:

* “Variational auto-encoders (VAEs) (Kingma & Welling, 2014) are DGMs trained by using mean-field
VI with a Gaussian parametric distribuion and amortization.” I disagree with this terminology—DGMs trained with, say, IAF are routinely called VAEs.

* Section 3.1: It might be good to clarify that you’re describing exact Hamiltonian integration, whereas in practice one always uses a discretized numerical integrator. (The leapfrog integrator is reversible and preserves volume, but doesn’t conserve energy, so this does make the results a bit more complicated.)

---

> ### Author Response · Authors · 2018-11-17
> **Thanks for the review and the hint of gap is in the paper**
>
> First, we would like to thank the reviewer's effort.
> "Here’s a concrete example of how I could imagine this procedure going wrong: make q(z0) a delta at the latent vector z* that maximizes the log-joint, and set the step size of the Hamiltonian simulation to 0. This will make the entropy term (which is ignored) -∞, maximize the log-joint term, and I think it even makes D^L_{KL}=0.
>
> It seems like this isn’t what actually happens experimentally, though—perhaps I’m missing something?"
>
> Perhaps we can clarify with the following points:
> 1) The irreducibility of ergodic Markov kernel (not in the paper)
> 2) HMC step size can never be 0 otherwise the kernel is not irreducible therefore not ergodic (not in the paper)
> 3) Our precondition (under eq(16))for dropping the entropy term (in the paper)
>
> First, a Markov chain is ergodic must satisfy following condition of irreducibility (precondition of recurrence): the probability of any state change to any other state must be positive.
>
> Second, if the simulation time/discretized step size in HMC is zero, then the state of Hamiltonian dynamics simply do not change. Therefore, the input and output of Hamiltonian simulation is the same with probability 1. This breaks the condition of ergodicity: irreducibility.
>
> Finally, we made it clear that there is a precondition for ignoring the entropy term: "If with the initial flow parameter φ0, the objective F˜ < Epθ(x,z)[log pθ(x, z)]..." (*right under* the definition of the loss without entropy (equation 16))
> We are happy to make this more connected with our motivation of ignoring the entropy term. It is not too hard to see why this precondition makes sense. In case of that we want to maximise this loss to converge to the target, then the initial loss value should be lower than the loss value of the target, otherwise maximising the loss will lead to diverge.
>
>
> On the comment:
> "Regarding the experiments, a natural baseline would be something akin to the approach of Hoffman (2017; "Learning Deep Latent Gaussian Models with Markov Chain Monte Carlo”),..."
> This is a fair point. We do have experiment results with this paper, but we didn't include it because Hoffman's method is much worse than VAE with the same amount of computation time. We are happy to add this results in, if the paper is accepted.

---

### Comment · AnonReviewer3 · 2018-10-25
**comparison to previous results**

The comparison to results from the literature might be slightly misleading. Please clarify.

* The toughest baseline that is compared to in section 4.2 is from 2015. Please include some more recent results for reference. There exist several better results.

* You write
"In (Salimans et al., 2015), the test likelihood is estimated using importence-weighted
samples from the encoder network. In our experiment, we use a more reliable estimation method
based on Hamiltonian annealled importance sampling and report the effective sample size (ESS)."
However, the method in the cited work is a lower bound and is thus conservative, whereas AIS is sometimes biased to the optimistic side. Can you clarify whether the reported numbers are indeed guaranteed to be comparable to the literature?

---

> ### Author Response · Authors · 2018-10-25
> **Some Clarifications on the Experiment of DeepConvNet on MNIST and Results (Section 4.2)**
>
> We are aware of that the baseline we chose is not the most recent work and there are better results in recent literature.
> We address the two comments as following.
>
> *First of all, this work is about a new inference method that can accelerate the training of deep generative models rather than new state-of-the-art models.
>
> For this purpose, the experiment in Section 4.2 is to demonstrate the training of deep generative models using our inference method over can converge faster than traditional variational inference in both wall-clock time and epochs.
> HVI (Salimans et al., 2015) is the most relevant baseline, because HVI is a variational method that uses HMC sampler as variational approximate distribution, that shares the same spirit of our method.
>
> Because the code of HVI is not publicly available, we reimplement their generator network and reproduced their results (that is in the middle rows of Table 1). For a fair comparison, we tested our method on the our implementation of the same generator network in  (Salimans et al., 2015). In our paper, we cited the results from Salimans et al. (that is in the top rows of Table 1) to support that our reimplementation (along with the estimation of log likelihood) is comparable to the original one in (Salimans et al., 2015).
>
> More recent works on variational inference for deep generative models are based on more advanced generator networks than the network used in (Salimans et al., 2015). There is no universal benchmark generator network that is used in most works. So, direct comparison with those results cross different works can be *misleading*, because it can be biased towards more advanced generator networks. To avoid this, we need to reimplement the generator networks used in other works, like what we did with HVI. Due to limited time, we could not do this. But, it is still good to add more recent works as a reference. We will do that in the next version of the paper.
>
> * In (Salimans et al., 2015), both lower bound and the test log likelihood are reported. The test log likelihood in (Salimans et al., 2015) is estimated by IS rather than AIS. In particular, Hamiltonian AIS we used is based on the recent work on "ON THE QUANTITATIVE ANALYSIS OF DECODER-BASED GENERATIVE MODELS" (Roger et al.,  ICLR 2017). We compare our method with the test log likelihood rather than the lower bound reported in (Salimans et al., 2015). The result of our HAIS evaluation on HVI is consistent with (Salimans et al., 2015). (Our reproduced result of HVI as shown in the middle rows of Table 1 is actually slightly worse than that in their paper) So, there is no direct evidence of the HAIS evaluation we used is biased. We are fairly confident that our results are comparable to (Salimans et al., 2015).

---

### Comment · AnonReviewer3 · 2018-10-30
**clarification on IS lower bound**

As a clarification to my earlier comment: The importance sampling estimator of the likelihood reported in the literature that is compared against is a lower bound (see the Importance Weighted Autoencoder paper by Burda et al), which the AIS method used in this paper is not. Not to be confused with the 1-sample variational lower bound that is also reported.

---

> ### Author Response · Authors · 2018-10-30
> **The difference of IS and AIS has been addressed in the submitted paper.**
>
> First, in your previous comment you mentioned explicitly lower bound rather than log likelihood as "However, the method in the cited work is a lower bound and is thus conservative, whereas AIS is sometimes biased to the optimistic side." This is why we emphasised that we reported the log likelihood rather than lower bound from the literature.
>
> Second, we are aware of the difference of IS and AIS log likelihood estimation as you commented here. We clarified this important detail In the submitted paper in the caption of Table 1: "... In (Salimans et al., 2015), the test likelihood is estimated using importence-weighted samples from the encoder network. In our experiment, we use a more reliable estimation method based on Hamiltonian annealled importance sampling ...".
>
> Moreover, we also report the ESS of our HAIS samplers. Effective sample size (ESS) is a well-known quality metric for the *all* kind of importance weighted sampling methods, like importence-weighted sampleing and Hamiltonian annealled importance sampling. ESS has been used in many literature for both IS and HAIS on evaluation of deep generative models, like "Approximate Inference with Amortised MCMC" from Li et al., icml, 2017 and "On the Quantitative Alysis of Decoder-based Generative Models", Grosse et al., ICLR, 2017.
>
> Finally, for more fair comparison with HVI, we have also reproduced the experiment of (Salimans et al., 2015) with HAIS evaluation which is reported in the middle rows of Table 1 (This has been mentioned in our previous comment.)

---

> ### Author Response · Authors · 2018-10-30
> **Clarification on IS and The relationship between IS and Annealed IS**
>
> *First, it is in standard textbook of Monte Carlo methods that *importance sampling gives unbiased estimation*.
> Given unnormalised density function p*(x), the normalisation constant is
> Z = \int p*(x)dx,
> which can be rewritten as
> Z = \int p*(x)/q(x) q(x) dx,
> where q(x) is some distribution.
> Using Monte Carlo method, we replace the integral with sampled average by samples from q(x)
> Z = \int p*(x)/q(x) q(x) dx
> \approx Z_{is}
> = 1/N \sum_i=1^N p*(x_i) / q(x_i),
> where x_i is sampled from q(x).
> The ratio w_i = p*(x_i) / q(x_i) is often known as the weight of samples.
> As a Monte Carlo estimator, Z_{is} is an unbiased estimator to Z independent of the number of samples  is 1 or 1000.
> However, the MC estimator can have a lower variance as the sample size N increases.
> Let Z_{is, k} = \sum_i=1^k w_i an k-sample unbiased estimator of Z. Z_{is, k} is essentially a random variable. Because it is unbiased estimator the mean of Z_{is, k} is equal to the true value Z, that is
> \log \Exp[Z_{is, k}] = \log \Exp[\sum_i=1^k w_i] = \log Z. (1)
> By Jensen's inequality, we have the lower bound
> \Exp[\log Z_{is, k}] \ge \log \Exp[Z_{is, k}].   (2)
> This is exactly the equation (9) in Importance Weighted Autoencoder paper by Burda et al.
>
> If we take k = 1, then the LHS of (2) above become 1-sample lower bound. Naturally, there are k-sample lower bound, that is different from k-sample IS estimator by (1).
> In (Salimans et al., 2015), the test log likelihood reported refers to the k-sample IS estimator (1) rather than k-sample lower bound (2).
>
> * Annealed IS ("Annealed Importance Sampling", Radford M. Neal, 1998) is simply a special IS method, where the q distribution is constructed as a sequence of annealing distributions q_1, q_2,... that systematically converges to p(x). The purpose of annealing distributions in AIS is to reduce the variance of weights *rather than* to be unbiased estimator.
>
> So, can you clarity the point of your comment? In particular, you said "The importance sampling estimator of the likelihood reported in the literature that is compared against is a lower bound (see the Importance Weighted Autoencoder paper by Burda et al), which the AIS method used in this paper is not."
>
> This difference should not be critical, because the quality of different IS estimators can be estimated by effective sample size.  For this reason, we report ESS for comparison of other work with different IS evaluation. But ESS is not reported in (Salimans et al., 2015).

---

> > ### Comment · AnonReviewer3 · 2018-10-30
> > **no**
> >
> > "In (Salimans et al., 2015), the test log likelihood reported refers to the k-sample IS estimator (1) rather than k-sample lower bound (2)."
> >
> > \log Z_{is, k} is what is reported in Salimans et al (for very large k). This is exactly the left hand side of your equation 2, which is indeed equation 9 in Burda et al. Your equation 1 includes intractable expectations (\Exp): How would Salimans et al. report those? Surely they just approximate by using Z_{is, k} and making k large? You're right that the reported number is meant to estimate Z, but the estimate is a lower bound (in expectation) and thus conservative. Pointing that out was one of the main contributions of Burda et al.
> >
> > "*First, it is in standard textbook of Monte Carlo methods that *importance sampling gives unbiased estimation*."
> >
> > Yes, Z_{is, k} is an unbiased estimator of Z, but \log Z_{is, k} is not unbiased, it's expectation is a lower bound.
> >
> > My point in making that comment was that if you claim that the IS based estimate is no good (inferior to AIS), it also means that all previous results in the literature are too conservative and that the comparison is biased in your favor.
> >
> > (* your equation 2 should have \leq where you have \ge)

---

> > > ### Author Response · Authors · 2018-10-30
> > > **Please read more basics on Importance sampling!**
> > >
> > > Please read "Importance Weighted Autoencoder" paper by Burda et al. carefully. In particularly, look at the equation (9).
> > >
> > > You said "Yes, Z_{is, k} is an unbiased estimator of Z, but \log Z_{is, k} is not unbiased, it is a lower bound.", but why?
> > >
> > > If in (Salimans et al., 2015), they made it clear that the they report test log likelihood by IS, that precisely means
> > > \log \Exp[Z_{is, k}] = \log \Exp[\sum_i=1^k w_i] = \log Z. (1)
> > > No one can argue this is wrong or not log likelihood. It is simple, the log of any unbiased estimator is also unbiased estimator.
> > >
> > > Not all log likelihood estimators are lower bound. Again all IS, AIS, HAIS are unbiased estimator. See more detailed explanation in "On the Quantitative Alysis of Decoder-based Generative Models", Grosse et al., ICLR, 2017.
> > >
> > > Can you explain why AIS is unbiased but IS is biased?

---

> > > > ### Comment · AnonReviewer3 · 2018-10-30
> > > > **no**
> > > >
> > > > "It is simple, the log of any unbiased estimator is also unbiased estimator."
> > > >
> > > > This is not true, and basic statistics.
> > > > I'm going to stop arguing now.

---

> > > > > ### Author Response · Authors · 2018-10-31
> > > > > **Appreciate your comments and Please let us know your concern**
> > > > >
> > > > > We appreciate the time and effort that you have invested in providing feedback about the paper and would like to apologize for any confusion regarding the previous responses.
> > > > >
> > > > > To clarify things, we would like to make the following points:
> > > > >
> > > > > 1) We agree with your comment that "Z_{is, k} is an unbiased estimator of Z, but \log Z_{is, k} is not unbiased, it's expectation is a lower bound.”.
> > > > >
> > > > > 2) We believe that you are concerned about the IS estimator having higher variance than the AIS estimator. If this were the case, when we take the log of the resulting estimate, the IS estimator could produce lower values on expectation than the AIS estimator.
> > > > >
> > > > > Our answer to 2) above is as follows:
> > > > >
> > > > > - The effective sample size (ESS) is a metric related to the variance of an IS estimator. To make a fair comparison with the literature, we reproduced the experiments in (Salimans et al., 2015) using HAIS and reported the ESS of the generated HAIS samples. In fact, the ESS metrics for our methods (Table 1, row 6-10) and HVI (Table 1, row 5) are similar (from 38 to 50). To this extent, we believe that comparing these different estimates of average test log marginal likelihood is fair because we obtain similar ESS metrics. We also agree that a direct comparison of our test log marginal likelihood metrics with other values reported in the literature that do not include ESS metrics can be unreliable (for example, if those other values could just be lower on average due to the higher variance of the used IS estimators).
> > > > >
> > > > > Please let us know if this has correctly addressed your concern or if your concern was actually different.

---

> > > > > > ### Comment · AnonReviewer3 · 2018-10-31
> > > > > > **thanks**
> > > > > >
> > > > > > Thanks, that's useful clarification, I agree.
> > > > > >
> > > > > > In addition to the bias of the IS estimator (= variance + log transform) my concern was also with the bias of AIS: Some results in some papers suggest that it may have an optimistic bias, see e.g. Figure 2 in Sohl-Dickstein & Culpepper, 2012, and that this bias may depend on the model, making comparison across models difficult. It's unclear to me in exactly what cases this is a problem and to what extent.
> > > > > >
> > > > > > My comment about the different likelihood estimators was only intended as a minor point: I believe your replication of the Salimans et al. result is a good effort towards a fair comparison.

---

### Meta-Review · Area_Chair1 · 2018-12-14
**issues remain to be addressed**

**Confidence:** 4
**Recommendation:** Reject

**Metareview:**

This paper proposes to a simple method for  tuning parameters of HMC by maximizing the log density under the final sample of the MCMC, and apply it for training VAE. The reviews and discussion raises some critical concerns and questions, which unfortunately, which unfortunately, is not adequately addressed.